# TRACE: Toulmin-based Reasoning Assessment through Constructive Elements for LLM CoT Evaluation

**Yundong Kim** [1 2]  **Heyoung Yang** [1]

## Abstract

Evaluating open-ended outputs from large language models (LLMs) remains challenging due to the absence of ground truth. Existing metrics rely on final-answer accuracy or surface-level statistics, leaving the reasoning process itself unexamined. We introduce TRACE (Toulmin-based Reasoning Assessment through Constructive Elements), a metric that analyzes Chain-of-Thought (CoT) reasoning processes. Rather than judging outcomes, TRACE inspects how arguments are constructed by integrating Toulmin's argumentation theory with Flavell's metacognitive framework to assess reasoning structure. Experiments on 26.3K QA samples across 7 reasoning models show strong correlation with benchmark accuracy ($r = 0.74$). Furthermore, TRACE is effective as a reinforcement learning reward signal, outperforming accuracy-only baselines. Together, these results indicate that logically sound reasoning leads to higher-quality answers. TRACE thus serves as a complementary metric for evaluating open-ended outputs. Code is available at https://github.com/hyyangkisti/trace.

## 1. Introduction

Large Language Models (LLMs) have demonstrated remarkable capabilities in complex problem-solving, largely driven by Chain-of-Thought (CoT) reasoning (Wei et al., 2022). By decomposing problems into intermediate steps, models can tackle tasks requiring multi-hop logic, mathematical deduction, and commonsense reasoning. However, as the complexity of these reasoning chains grows, evaluating their

[1]Applied Agent Research Center, Korea Institute of Science and Technology Information (KISTI), Republic of Korea [2]Department of Computer Science and Engineering, University of Seoul, Republic of Korea. Correspondence to: Heyoung Yang <hyyang@kisti.re.kr>.

*Proceedings of the 43rd International Conference on Machine Learning*, Seoul, South Korea. PMLR 306, 2026. Copyright 2026 by the author(s).

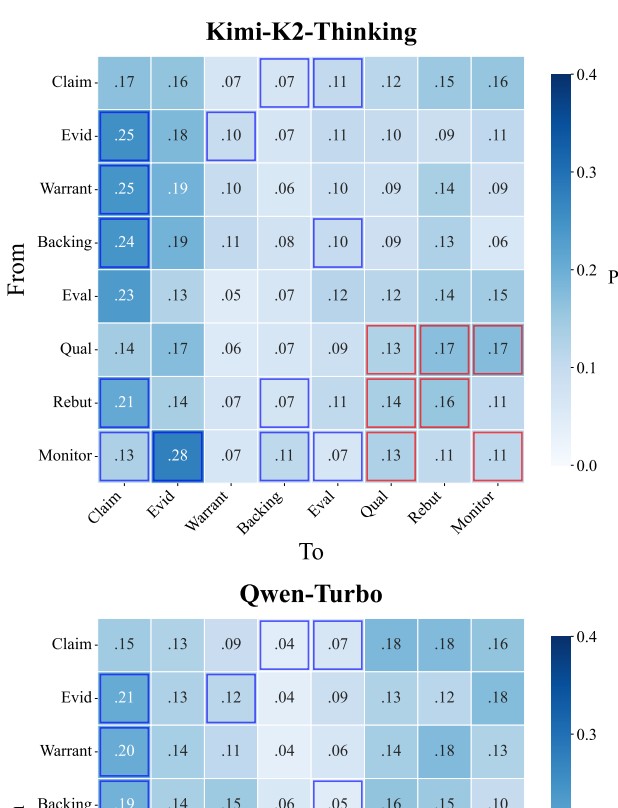

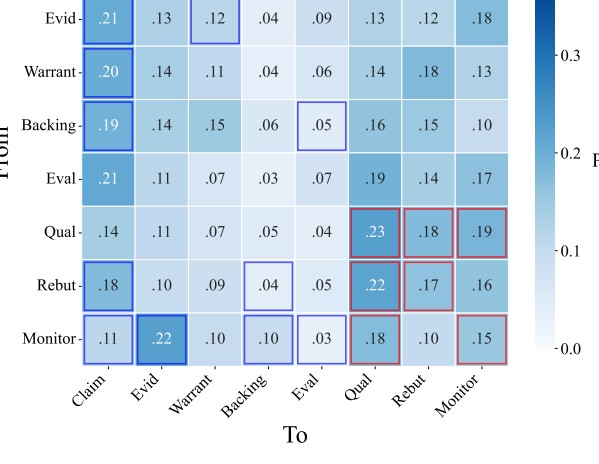

*Figure 1.* Transition heatmaps comparing Kimi-K2-Thinking and Qwen-Turbo. Blue-bordered cells denote Good Transitions (e.g., Evidence → Claim); Red-bordered cells denote Bad Transitions (e.g., Monitoring → Qualifier).

quality becomes increasingly challenging. Current evaluation paradigms focus on outcome-based metrics (e.g., accuracy, exact match), which assess the final answer but treat the reasoning process as a black box. Reference-free metrics,

including Perplexity (Radford et al., 2019), Token Length, and Lexical Diversity (MTLD) (McCarthy & Jarvis, 2010), offer insights into statistical fluency or diversity. However, it is difficult to capture whether an LLM engages in genuine logical reasoning relying solely on these metrics.

Consequently, there is a growing need for process-based evaluation that can diagnose how a model thinks, not just what it concludes. Although "LLM-as-a-judge" approaches (Zheng et al., 2023) have gained popularity, they are often limited to relative evaluations (e.g., A/B testing) and remain black box in their decision making process, making it difficult to pinpoint specific reasoning flaws. To address these limitations, this study aims to quantify the quality of LLM reasoning by grounding it in established argumentation theory and cognitive science.

In this paper, we propose **TRACE** (**T**oulmin-based **R**easoning **A**ssessment through **C**onstructive **E**lements), a novel reference-free framework for evaluating LLM CoT quality based on Toulmin's Argumentation Model (Toulmin, 2003) and Flavell's Metacognition Theory (Flavell, 1979). Toulmin's core abstraction of claim, data, and warrant is domain-agnostic and naturally fits the think-aloud nature of CoT reasoning. As illustrated in the Transition Heatmaps in Figure 1, we operationalize this by decomposing LLM reasoning blocks into sentence-level units and employing TRACE-DeBERTa to multi-label the Constructive Elements inherent in each sentence. These heatmaps quantify the reasoning flow by visualizing the transition probabilities from one element to the next. We classify these shifts based on a predefined Transition Set grounded in Toulmin's and Flavell's frameworks.

Blue-bordered cells represent 'Good Transitions' (e.g., Evidence $\rightarrow$ Claim), indicating robust structural integrity, whereas red-bordered cells represent 'Bad Transitions' (e.g., Monitoring $\rightarrow$ Qualifier), highlighting areas of cognitive confusion. Empirically, we observe that Kimi-k2-thinking exhibits a higher frequency of Good Transitions and significantly fewer Bad Transitions compared to Qwen-Turbo. This distinction validates our hypothesis that while logical progression correlates with correctness, excessive hesitation often serves as a proxy for reasoning uncertainty rather than effective self regulation.

Based on these Constructive Elements, we compute a TRACE Score, a composite metric derived from two components: **(1) State Validity**, which assesses the validity of individual reasoning steps based on allowed constructive states, and **(2) Transition Coherence**, which evaluates the logical flow between steps using a transition matrix.

We validated TRACE through three distinct experiments. First, we analyzed the correlation between TRACE Scores and model accuracy across 39 established benchmarks (e.g.,

MMLU, GPQA) using 7 prominent LLMs, utilizing 26.3K pairs of reasoning blocks. Results demonstrate that TRACE achieves a strong Pearson correlation of ($r = 0.741$) with accuracy. Second, we assessed the alignment between TRACE and LLM-as-a-Judge using Arena-Hard-v2.0 (Li et al., 2025), achieving a 64% agreement rate in the MATH category. Third, we demonstrated the practical utility of TRACE as a Reinforcement Learning reward signal on GSM8K, where it yielded superior performance improvements (+9.9% over the base model) compared to using accuracy-based rewards alone.

## 2. Related Work

**QA Benchmarks and CoT Evaluation** Standard LLM evaluation has long relied on static Question-Answering (QA) benchmarks like MMLU and GPQA. However, as model capabilities saturate these metrics, recent initiatives such as *LiveBench* (White et al., 2025) and *Humanity's Last Exam* (Phan et al., 2025) have introduced frontier-level problems to enhance discriminative power. Despite their increased difficulty, these benchmarks remain fundamentally outcome-based, relying on ground-truth labels (e.g., multiple-choice or short-answer) that overlook the quality of the underlying Chain-of-Thought (CoT) process. To address this, studies like *MR-GSM8K* (Zeng et al., 2025), *CofCA* (Wu et al., 2025), *ProcessBench* (Zheng et al., 2025a) and *PRM* (Khalifa et al., 2025) have proposed decomposing reasoning into intermediate steps for finer-grained evaluation. However, these methods often require step-level correctness labels or heavyweight verifier models, limiting their scalability. In contrast, TRACE is designed to be fully reference-free and lightweight, enabling efficient evaluation without ground-truth supervision or expensive inference.

**LLM-as-a-Judge** To overcome the rigidity of QA metrics, the "LLM-as-a-judge" paradigm has gained prominence, employing strong models (e.g., GPT-4) as evaluators. Benchmarks like *FLASK* (Ye et al., 2024), *AlpacaEval* (Dubois et al., 2025), *WildBench* (Lin et al., 2025), and *Arena-Hard-v2.0* utilize pairwise comparisons to approximate human preference, showing high correlation with the *LMSYS Chatbot Arena* (Chiang et al., 2024). Meanwhile, works like *MT-Bench-101* (Bai et al., 2024), *JudgeLRM* (Chen et al., 2025) and *HelpSteer2* (Wang et al., 2024b) extend this to multi-turn dialogues using hierarchical capability scoring rather than simple A/B testing. However, recent research points to potential limitations within this paradigm. Surveys (Chen et al., 2024; Gu et al., 2026; Tan et al., 2025) indicate that LLM judges may occasionally exhibit biases, such as a preference for verbosity or specific positioning. Notably, (Zheng et al., 2025b) observed that models could sometimes achieve high win rates by aligning with these stylistic preferences, even without robust reasoning capabil-

ities. While recent judge models provide fine-grained rubric scores, engineers often lack visibility into which specific reasoning steps contributed to each score, making it difficult to trace and remediate particular weaknesses.

**Argumentation Mining**  Early research in this field focused on identifying functional units, such as claims and premises, within human persuasive essays (Stab & Gurevych, 2014). We adapt this established paradigm to the domain of Large Language Models, shifting the focus from human writing to machine-generated Chain-of-Thought (CoT) processes.

## 3. Methodology

The TRACE framework employs a two-stage pipeline to quantify LLM reasoning quality. First, reasoning blocks are segmented into sentences via spaCy and classified by TRACE-DeBERTa to identify constructive attributes. Second, we evaluate the State Validity and Transition Coherence of the resulting label sequence through a rule-based algorithm. This yields a metric reflecting both logical integrity and cognitive dissonance.

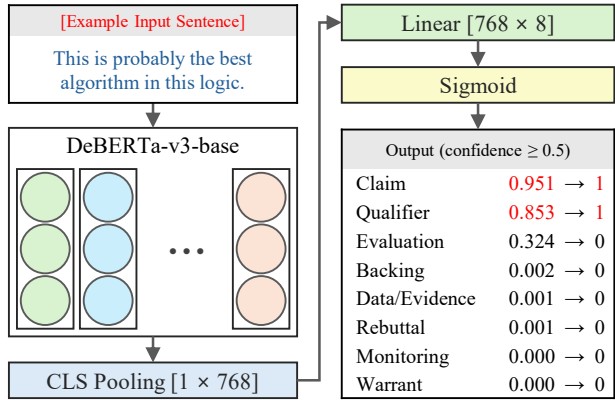

*Figure 2.* Architecture of TRACE-DeBERTa. The model encodes an input reasoning sentence using DeBERTa-v3-base. The [CLS] representation is projected to an 8-dimensional confidence vector via a linear layer and Sigmoid activation, enabling multi-label classification of constructive elements.

### 3.1. TRACE-DeBERTa for Sentence Attributes Labeling

**Model Selection**  We select DeBERTa-v3-base (He et al., 2023) as our backbone. While RoBERTa (Liu et al., 2019) remains a strong baseline, we opted for the more recent DeBERTa architecture for its disentangled attention mechanism. We also considered ModernBERT (Warner et al., 2025), which excels in long-context efficiency. However, (Antoun et al., 2025) reports that DeBERTa retains a slight edge on short-sequence classification tasks. Given that our framework operates on sentence-level inputs, we prioritize DeBERTa for its proven precision in fine-grained classifica-

tion.

**Architecture and Data**  Inspired by the GoEmotions framework (Demszky et al., 2020), TRACE-DeBERTa employs a fine-grained multi-label classification head. As shown in Figure 2, the pooled [CLS] embedding is mapped to an 8-dimensional logit vector via a linear layer, followed by sigmoid activation to produce label probabilities. We apply a sigmoid activation and adopt labels with probability $\geq 0.5$.

The model was fine-tuned using BCEWithLogitsLoss with class-specific weights to address label imbalance, where weights were softened by interpolating between uniform and frequency-based values. Training data consisted of ~100k reasoning sentences annotated by advanced LLMs (GPT-5.1 and Claude 4.5 Sonnet, alternated to mitigate single-model stylistic bias) using few-shot prompts grounded in Toulmin's and Flavell's definitions. More details, see Appendix B.1

*Table 1.* TRACE-DeBERTa performance evaluated against human annotations (400 sentences stratified across models and label categories, 3 senior NLP researchers, inter-annotator agreement Cohen's $\kappa = 0.672$).

| Label | Precision | Recall | F1-score |
|---|---|---|---|
| Claim | 0.696 | 0.634 | 0.662 |
| Data/Evidence | 0.774 | 0.588 | 0.663 |
| Warrant | 0.602 | 0.544 | 0.547 |
| Backing | 0.780 | 0.612 | 0.685 |
| Qualifier | 0.865 | 0.783 | 0.821 |
| Rebuttal | 0.712 | 0.549 | 0.619 |
| Monitoring | 0.803 | 0.585 | 0.675 |
| Evaluation | 0.610 | 0.711 | 0.654 |
| **Macro Avg** | **0.730** | **0.626** | **0.666** |

**Performance**  To independently validate the classifier, three senior NLP researchers annotated 400 sentences stratified across models and label categories. Inter-annotator agreement reached Cohen's $\kappa = 0.672$, reflecting the inherent difficulty of this fine-grained task. Against these human labels, TRACE-DeBERTa achieves a Macro F1-score of 0.666 (Table 1). This approaches the inter-annotator agreement ceiling, suggesting that remaining errors largely reflect task ambiguity rather than systematic failure. Per-category performance varies with the explicitness of surface markers. *Qualifier* (F1 = 0.821) scores highest, since its linguistic cues are clear. *Warrant* (F1 = 0.547) scores lowest, since it captures implicit inferential links that overlap with adjacent categories. Overall, the classifier is sufficiently reliable for downstream TRACE score computation.

### 3.2. TRACE Score Extraction

Once the reasoning block is converted into a sequence of label sets $L = \{l_1, l_2, \ldots, l_n\}$ by TRACE-DeBERTa, we

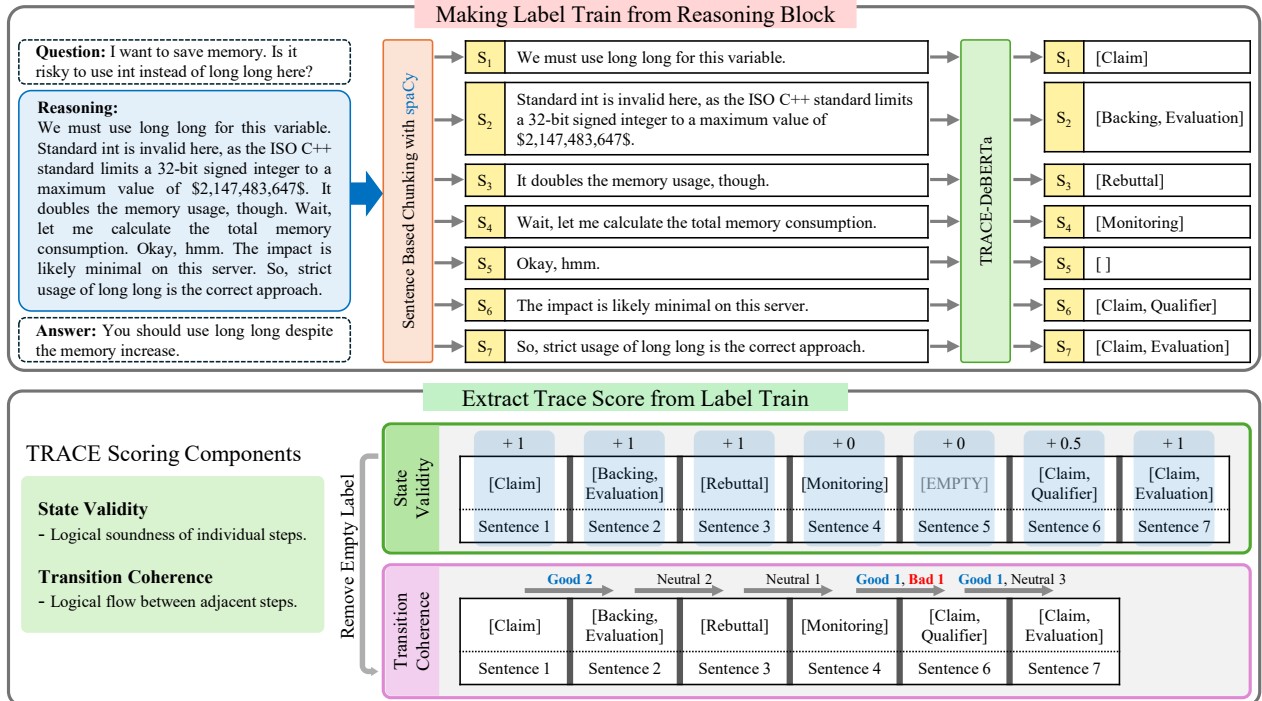

*Figure 3.* **Overview of the TRACE Pipeline.** The framework operates in two main phases: (Top) Making Label Train from Reasoning Block, where the raw reasoning text is decomposed and multi-labeled by TRACE-DeBERTa (described in Figure 2); and (Bottom) Extract TRACE Score from Label Train, where the resulting label sequence is analyzed for State Validity and Transition Coherence to compute the final metric.

proceed to the second phase illustrated in Figure 3: Extract Trace Score from Label Train. The scoring mechanism captures both the static structural integrity of individual steps and the dynamic logical flow of the argument. The final TRACE value is defined as a weighted sum of **State Validity** and **Transition Coherence**:

$$\text{TRACE} = \alpha \cdot V_{state} + (1 - \alpha) \cdot C_{trans} \qquad (1)$$

We assign higher weight to State Validity ($\alpha = 0.7$) based on the principle that local coherence precedes global coherence. In Toulmin's framework, a valid argument must first establish well-formed units (e.g., Claim, Data, or their valid combinations) before chaining them into extended reasoning. If individual sentences lack valid argumentative structure, the quality of their sequential flow becomes irrelevant. See Appendix C.1 for empirical validation.

**State Validity** ($V_{state}$) State Validity evaluates whether each sentence forms a logically sound argument unit. We define a set of Allowed States ($\mathcal{S}_{allowed}$), derived from valid combinations of Toulmin's components (e.g., Claim, Data, Warrant), while penalizing ambiguous or structurally weak combinations. For each sentence $i$ with label set $l_i$, we calculate the Jaccard Similarity $J(l_i, s)$ against the allowed states:

$$V_{state} = \frac{1}{N} \sum_{i=1}^{N} \max\{J(l_i, s) \mid s \in \mathcal{S}_{allowed}\} \qquad (2)$$

To accommodate the diverse nature of reasoning, $\mathcal{S}_{allowed}$ includes both single and composite attributes. We define the set as follows:

$$\mathcal{S}_{allowed} = \{\text{'Claim', 'Data', 'Warrant',} \\ \text{'Backing', 'Backing+Evaluation', \dots}\} \qquad (3)$$

For instance, a distinct logical statement like *'Backing+Evaluation'* ($J = 1.0$) contributes fully to validity, whereas a hedged statement like *'Qualifier+Claim'* ($J = 0.5$) yields a lower score, reflecting structural uncertainty. Consequently, this mechanism inherently penalizes non-constructive attributes such as isolated *Monitoring* or excessive *Qualifiers* by assigning them lower validity scores compared to robust argumentative components.

**Transition Coherence** ($C_{trans}$) Transition Coherence assesses the logical quality of the flow between valid reasoning steps. Prior to analysis, we filter out sentences with no assigned labels (EMPTY), as these typically represent phatic expressions or fillers (e.g., "Let's see") that introduce noise without contributing to the argumentative structure. For the remaining sequence of non-empty label sets, we evaluate ev-

*Table 2.* Accuracy and TRACE scores across 7 LLMs on 39 benchmarks. Top-2 values per AVERAGE row are in **bold**.

| DATASET | GPT-OSS 120B | | GPT-OSS 20B | | Claude 3.7 Sonnet | | Qwen Turbo | | Qwen Flash | | DeepSeek R1 | | Kimi K2 Thinking | |
|---|---|---|---|---|---|---|---|---|---|---|---|---|---|---|
| (max samples per data ≤ 100) | Acc | TRACE | Acc | TRACE | Acc | TRACE | Acc | TRACE | Acc | TRACE | Acc | TRACE | Acc | TRACE |
| **AIME** | | | | | | | | | | | | | | |
| aime24 | 77% | 0.644 | 67% | 0.638 | 37% | 0.590 | 80% | 0.564 | 77% | 0.586 | 93% | 0.605 | 90% | 0.628 |
| aime25 | 87% | 0.638 | 63% | 0.608 | 27% | 0.573 | 53% | 0.554 | 67% | 0.579 | 90% | 0.556 | 80% | 0.627 |
| AVERAGE | 82% | **0.641** | 65% | 0.623 | 32% | 0.582 | 67% | 0.559 | 72% | 0.583 | 92% | 0.581 | 85% | **0.628** |
| **GSM8K** | **99%** | **0.751** | **98%** | 0.686 | 95% | **0.701** | **99%** | 0.620 | **99%** | 0.636 | 97% | 0.591 | **98%** | 0.646 |
| **ARC** | | | | | | | | | | | | | | |
| arc-easy | 100% | 0.726 | 99% | 0.616 | 99% | 0.688 | 98% | 0.567 | 98% | 0.632 | 100% | 0.645 | 82% | 0.676 |
| arc-challenge | 96% | 0.695 | 92% | 0.614 | 97% | 0.669 | 95% | 0.550 | 92% | 0.626 | 94% | 0.635 | 79% | 0.667 |
| AVERAGE | **98%** | **0.711** | 96% | 0.615 | **98%** | **0.679** | 97% | 0.559 | 95% | 0.629 | **97%** | 0.640 | 81% | 0.672 |
| **MMLU** | | | | | | | | | | | | | | |
| college biology | 96% | 0.724 | 95% | 0.610 | 99% | 0.701 | 90% | 0.585 | 96% | 0.625 | 98% | 0.649 | 93% | 0.669 |
| college chemistry | 77% | 0.664 | 77% | 0.619 | 76% | 0.666 | 64% | 0.563 | 71% | 0.566 | 80% | 0.595 | 76% | 0.640 |
| college computer science | 97% | 0.669 | 94% | 0.631 | 95% | 0.674 | 70% | 0.577 | 75% | 0.585 | 98% | 0.605 | 96% | 0.675 |
| college mathematics | 97% | 0.693 | 98% | 0.651 | 95% | 0.623 | 49% | 0.572 | 65% | 0.604 | 98% | 0.565 | 95% | 0.641 |
| college medicine | 90% | 0.674 | 86% | 0.616 | 86% | 0.666 | 86% | 0.558 | 89% | 0.599 | 89% | 0.611 | 85% | 0.650 |
| college physics | 99% | 0.773 | 99% | 0.689 | 98% | 0.720 | 79% | 0.628 | 77% | 0.618 | 100% | 0.626 | 97% | 0.685 |
| AVERAGE | **93%** | **0.700** | 92% | 0.636 | 92% | **0.675** | 73% | 0.581 | 79% | 0.600 | **94%** | 0.609 | 90% | 0.660 |
| **MMLU-PRO** | | | | | | | | | | | | | | |
| biology | 86% | 0.618 | 77% | 0.609 | 89% | 0.677 | 87% | 0.588 | 85% | 0.604 | 85% | 0.612 | 90% | 0.638 |
| business | 81% | 0.661 | 78% | 0.592 | 84% | 0.606 | 82% | 0.583 | 81% | 0.571 | 73% | 0.571 | 87% | 0.604 |
| chemistry | 81% | 0.630 | 80% | 0.618 | 89% | 0.640 | 83% | 0.566 | 87% | 0.611 | 81% | 0.577 | 89% | 0.627 |
| computer science | 81% | 0.583 | 79% | 0.589 | 84% | 0.649 | 83% | 0.565 | 83% | 0.590 | 78% | 0.588 | 89% | 0.629 |
| economics | 82% | 0.612 | 72% | 0.605 | 86% | 0.652 | 80% | 0.589 | 86% | 0.603 | 80% | 0.591 | 87% | 0.632 |
| engineering | 71% | 0.618 | 63% | 0.586 | 71% | 0.605 | 71% | 0.546 | 74% | 0.590 | 65% | 0.564 | 84% | 0.612 |
| health | 74% | 0.576 | 65% | 0.578 | 66% | 0.605 | 72% | 0.526 | 72% | 0.560 | 76% | 0.551 | 79% | 0.583 |
| history | 64% | 0.565 | 58% | 0.583 | 73% | 0.574 | 65% | 0.523 | 65% | 0.569 | 69% | 0.591 | 73% | 0.597 |
| law | 57% | 0.569 | 39% | 0.602 | 66% | 0.618 | 51% | 0.554 | 58% | 0.587 | 68% | 0.579 | 69% | 0.634 |
| math | 92% | 0.651 | 92% | 0.643 | 90% | 0.653 | 92% | 0.623 | 92% | 0.637 | 84% | 0.595 | 95% | 0.661 |
| other | 59% | 0.565 | 52% | 0.561 | 76% | 0.604 | 53% | 0.507 | 64% | 0.547 | 71% | 0.541 | 77% | 0.599 |
| philosophy | 68% | 0.523 | 64% | 0.537 | 81% | 0.588 | 71% | 0.524 | 71% | 0.554 | 77% | 0.546 | 84% | 0.588 |
| physics | 86% | 0.649 | 86% | 0.616 | 84% | 0.650 | 86% | 0.582 | 93% | 0.617 | 77% | 0.587 | 90% | 0.639 |
| psychology | 76% | 0.575 | 70% | 0.590 | 73% | 0.612 | 71% | 0.548 | 75% | 0.564 | 78% | 0.579 | 79% | 0.618 |
| AVERAGE | 76% | 0.600 | 70% | 0.594 | **79%** | **0.624** | 75% | 0.559 | 78% | 0.586 | 76% | 0.577 | **84%** | **0.619** |
| **GPQA** | | | | | | | | | | | | | | |
| gpqa main | 67% | 0.591 | 59% | 0.563 | 64% | 0.613 | 38% | 0.483 | 50% | 0.541 | 75% | 0.589 | 72% | 0.600 |
| gpqa diamond | 67% | 0.586 | 63% | 0.556 | 72% | 0.606 | 36% | 0.485 | 57% | 0.538 | 79% | 0.571 | 74% | 0.610 |
| AVERAGE | 67% | 0.589 | 61% | 0.560 | 68% | **0.610** | 37% | 0.484 | 54% | 0.540 | **77%** | 0.580 | **73%** | **0.605** |
| **SuperGPQA** | | | | | | | | | | | | | | |
| agronomy | 40% | 0.563 | 34% | 0.528 | 46% | 0.572 | 44% | 0.460 | 42% | 0.501 | 47% | 0.509 | 53% | 0.556 |
| economics | 39% | 0.607 | 33% | 0.569 | 50% | 0.605 | 51% | 0.500 | 57% | 0.541 | 60% | 0.546 | 61% | 0.603 |
| education | 48% | 0.566 | 39% | 0.511 | 54% | 0.573 | 53% | 0.458 | 51% | 0.510 | 49% | 0.533 | 57% | 0.582 |
| engineering | 56% | 0.599 | 46% | 0.571 | 60% | 0.593 | 58% | 0.494 | 59% | 0.525 | 67% | 0.543 | 70% | 0.587 |
| history | 35% | 0.508 | 26% | 0.481 | 57% | 0.531 | 40% | 0.409 | 40% | 0.442 | 56% | 0.508 | 53% | 0.567 |
| law | 42% | 0.585 | 32% | 0.555 | 54% | 0.582 | 47% | 0.495 | 49% | 0.537 | 71% | 0.538 | 73% | 0.608 |
| literature & arts | 41% | 0.497 | 26% | 0.487 | 57% | 0.553 | 38% | 0.437 | 40% | 0.463 | 56% | 0.507 | 63% | 0.564 |
| management | 55% | 0.560 | 31% | 0.529 | 59% | 0.567 | 43% | 0.461 | 48% | 0.507 | 57% | 0.514 | 56% | 0.562 |
| medicine | 48% | 0.595 | 40% | 0.560 | 53% | 0.578 | 56% | 0.464 | 62% | 0.541 | 57% | 0.532 | 61% | 0.596 |
| philosophy | 50% | 0.571 | 41% | 0.538 | 60% | 0.585 | 51% | 0.485 | 55% | 0.527 | 71% | 0.552 | 59% | 0.586 |
| science | 69% | 0.611 | 70% | 0.599 | 68% | 0.586 | 68% | 0.518 | 69% | 0.554 | 71% | 0.541 | 77% | 0.592 |
| sociology | 38% | 0.557 | 35% | 0.511 | 59% | 0.567 | 38% | 0.431 | 47% | 0.485 | 63% | 0.513 | 58% | 0.593 |
| AVERAGE | 47% | 0.568 | 38% | 0.537 | 56% | **0.574** | 49% | 0.468 | 52% | 0.511 | **60%** | 0.528 | **62%** | **0.583** |

ery adjacent pair $(l_i, l_{i+1})$. We define a set of **Good Transitions** ($\mathcal{T}_{good}$), representing robust logical progressions (e.g., *Data → Claim*), and **Bad Transitions** ($\mathcal{T}_{bad}$), indicating cognitive stalling or circular uncertainty (e.g., *Monitoring → Qualifier*). Any transition not explicitly categorized in $\mathcal{T}_{good}$ or $\mathcal{T}_{bad}$ is treated as a Neutral Transition.

The classification of transitions is grounded in the theoretical roles of each element. In Toulmin's framework, *Data*, *Warrant*, and *Backing* serve as support-providing elements that should flow toward *Claim*. Thus, transitions such as

*Data → Claim* or *Warrant → Claim* represent the natural completion of an argumentative unit. Conversely, *Monitoring* and *Qualifier* signal uncertainty or self-regulation in Flavell's metacognitive framework. When these elements follow each other (e.g., *Monitoring → Qualifier*), the reasoning process accumulates hesitation without resolution.

The coherence score is normalized to $[0, 1]$ based on the net density of positive transitions:

$$C_{trans} = \frac{1}{2} \left( \frac{N_{good} - N_{bad}}{N_{total}} + 1 \right) \quad (4)$$

where $N_{total}$ represents the total number of transitions in the reasoning block ($N_{total} = N_{good} + N_{bad} + N_{neutral}$). This normalization ensures that the metric reflects the density of valid logical progressions relative to the overall length of the chain.

These configurations were optimized by empirically validating various permutations grounded in Toulmin's and Flavell's theories. The full definitions of $\mathcal{S}_{allowed}$, $\mathcal{T}_{good}$, and $\mathcal{T}_{bad}$ are provided in Appendix B.2 and Appendix B.3.

## 4. Experiments

We validate TRACE through three experiments. Experiment 1 examines the correlation between TRACE scores and ground-truth accuracy across standard benchmarks. Experiment 2 assesses the alignment with "LLM-as-a-judge" preferences on Arena Hard v2.0. Experiment 3 demonstrates the practical utility of TRACE as a reward signal for Reinforcement Learning (RL) to enhance model reasoning capabilities.

### 4.1. Experiment 1: Correlation with Benchmark Accuracy

**Setup**  We utilized 39 widely-used benchmarks covering diverse domains (Math, Science, Coding, Humanities, etc.), including AIME, GSM8K (Cobbe et al., 2021), ARC (Clark et al., 2018), MMLU (Hendrycks et al., 2021), MMLU-PRO (Wang et al., 2024a), GPQA (Rein et al., 2024) and SuperG-PQA (Du et al., 2025). We evaluated 7 primary LLMs: GPT-oss-120b, GPT-oss-20b (Agarwal et al., 2025), Claude-3.7-Sonnet-20250219 (Anthropic, 2024), Qwen-Turbo (Qwen et al., 2025), Qwen-Flash (Yang et al., 2025), Deepseek-R1-0528 (Guo et al., 2025), and Kimi-K2-Thinking (Team et al., 2025). Since most proprietary models do not expose their intermediate reasoning processes, our analysis predominantly employs open-source models. To ensure diversity over depth, we sampled up to 100 instances per dataset, resulting in a total of 26,320 reasoning samples. While this represents a subset, recent studies (Kipnis et al., 2025) suggest that sparse sampling provides a reasonable proxy for assessing overall model capabilities. We standardized evaluations using *lm-evaluation-harness* (Biderman et al., 2024).

**Main Results**  We calculated the accuracy for each benchmark and the mean TRACE value of the generated reasoning blocks. Table 2 presents the detailed breakdown, and Figure 4 visualizes a clear positive linear trend between TRACE and accuracy.

Table 3 compares the Pearson ($r$) and Spearman ($\rho$) correlations of TRACE against baseline metrics. TRACE achieves a Pearson correlation of +0.741, drastically outperforming

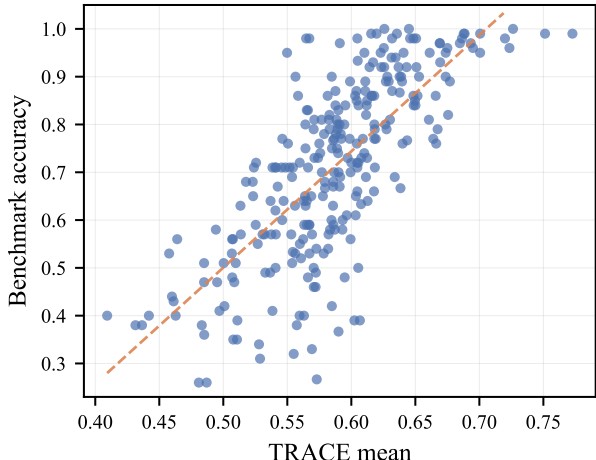

*Figure 4.* Scatter plot of benchmark accuracy versus mean TRACE score across all model-benchmark pairs ($n = 273$).

surface-level metrics such as Token Length ($r = -0.147$), Perplexity ($r = +0.221$) and MTLD ($r = -0.207$). While extended reasoning typically correlates with improved performance within a controlled setting (i.e., a fixed model and task), raw token length becomes unreliable when comparing across heterogeneous models and benchmarks. Similarly, lexical diversity and fluency show limited predictive power, suggesting that the structural validity captured by TRACE is a far better predictor of correctness than statistical text properties.

*Table 3.* Correlation between evaluation metrics and benchmark accuracy.

| Method | Pearson $r$ | Spearman $\rho$ |
|---|---|---|
| Token Length | $-0.147$ | $-0.186$ |
| Perplexity | $+0.221$ | $+0.145$ |
| MTLD | $-0.207$ | $-0.244$ |
| **TRACE (Ours)** | $\mathbf{+0.741}$ | $\mathbf{+0.755}$ |

Furthermore, as shown in Table 4, TRACE maintains strong correlation ($r > 0.78$) across all 7 models. This robust intra-model correlation suggests that TRACE can be effective for model alignment—maximizing TRACE within a model's native generation style may help unlock its peak reasoning potential. This per-model view shows that, with model identity controlled, the correlation observed in aggregate also holds within each individual model. A similar pattern appears within the same model family. In Table 2, GPT-OSS-120B yields higher mean TRACE scores than GPT-OSS-20B across the benchmark groups. This is consistent with the expected effect of model scale.

*Table 4.* per-model correlation between TRACE mean and benchmark accuracy.

| LLM | Pearson $r$ | $n$ Benchmarks |
|---|---|---|
| gpt-oss-120b | 0.8216 | 39 |
| gpt-oss-20b | 0.8555 | 39 |
| claude-3.7-sonnet | 0.8106 | 39 |
| qwen-turbo | 0.8179 | 39 |
| qwen-flash | 0.9144 | 39 |
| deepseek-r1 | 0.8340 | 39 |
| kimi-k2-thinking | 0.7829 | 39 |

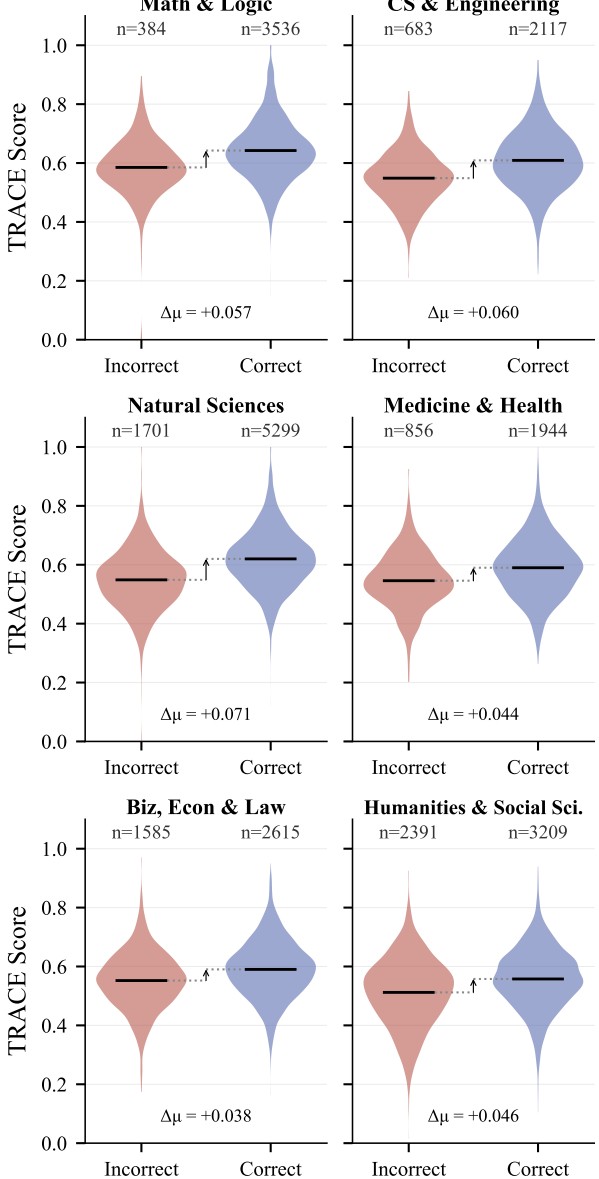

*Figure 5.* TRACE score distributions for correct vs. incorrect answers by domain.

**Domain Analysis** To investigate how reasoning structure impacts performance across different fields, we clustered the 39 benchmarks into 6 categories: Math & Logic, CS & Engineering, Natural Sciences, Medicine & Health, Biz/Econ/Law, and Humanities & Social Sci. The detailed mapping is provided in Appendix A.4. We generated violin plots Figure 5 to visualize the distribution of TRACE values for Correct vs. Incorrect answers.

We observed that for correct answers, the TRACE value is consistently higher than for incorrect ones. We quantified this by calculating the difference in means ($\Delta\mu = \mu_{correct} - \mu_{incorrect}$):

- Math & Logic: $\Delta\mu = +0.057$
- CS & Engineering: $\Delta\mu = +0.060$
- Natural Sciences: $\Delta\mu = +0.071$ **(Highest)**
- Medicine & Health: $\Delta\mu = +0.044$
- Humanities & Social Sci.: $\Delta\mu = +0.046$
- Biz, Econ & Law: $\Delta\mu = +0.038$ **(Lowest)**

The results indicate that in logic-intensive domains like Natural Sciences and Math, the gap is most pronounced, suggesting that tasks requiring rigorous deductive steps are more sensitive to argumentative structure. In contrast, domains relying more on knowledge retrieval (Biz/Law) show a smaller, yet still positive, gap. With domain controlled as a variable, TRACE separates correct from incorrect answers across all six categories, with a larger gap in deductive domains and a smaller gap in knowledge-retrieval domains.

### 4.2. Experiment 2: Alignment with LLM-as-a-judge

**Setup** To evaluate TRACE in an open-ended generation setting where ground truth is not always binary, we used the Arena Hard v2.0 benchmark (English subset). We pitted DeepSeek-R1 against QwQ-32b and used GPT-4.1 as the judge to determine the "Winner." We then assessed whether TRACE (and baselines) could correctly predict the winner by assigning the win to the model with the higher metric score.

*Table 5.* Performance comparison of different metrics in prediction accuracy against GPT-4.1 judge on Arena Hard v2.0 (EN).

| Method | coding | creative writing | math |
|---|---|---|---|
| Token Length | 44.22% | 47.85% | 36.78% |
| Perplexity | 49.21% | 47.79% | 50.57% |
| MTLD | 52.96% | **53.45%** | 58.62% |
| **TRACE (Ours)** | **55.78%** | 50.00% | **64.37%** |

**Results** Table 5 summarizes the prediction accuracy of each metric against the GPT-4.1 judge decisions. In Math tasks, TRACE achieved a prediction accuracy of 64.37%, outperforming heuristic baselines. In Coding, TRACE also led with 55.78%. However, for Creative Writing, MTLD proved to be a more effective predictor 53.45%, which aligns

with the expectation that creative tasks prioritize lexical diversity over argumentative structure.

While these results show improvements over baselines, the overall prediction accuracies remain moderate. We attribute this to two primary factors. First, TRACE is designed to detect argumentative structure and cognitive dissonance, not factual correctness; a model may reason fluently yet produce erroneous intermediate steps or conclusions, leading to disagreement with LLM judge decisions. Second, the current pipeline processes all textual content uniformly, including code blocks and narrative segments. These non-argumentative elements introduce noise into the constructive element classification, as TRACE-DeBERTa was trained predominantly on natural language reasoning and LaTeX equations, resulting in limited robustness to inputs dominated by raw code or creative narrative.

Nevertheless, the performance in the Math category aligns with the design of TRACE, which focuses on deductive reasoning structures. This suggests that the metric provides a relevant signal in reasoning-intensive domains. It also remains lightweight relative to LLM-judge approaches (Appendix A.3), making it easy to apply as a quick diagnostic.

## 4.3. Experiment 3: TRACE as a Reinforcement Learning Reward Signal

**Setup** We fine-tuned the DeepSeek-R1-Distill-Qwen-1.5B model using Group Relative Policy Optimization (GRPO) (Shao et al., 2024). To isolate the benefit of structural guidance, we defined a composite reward function $R(o, y^*)$ as:

$$R(o, y^*) = \lambda_{\text{acc}} \cdot r_{\text{acc}}(o, y^*) + r_{\text{trace}}(o) + r_{\text{len}}(o) \quad (5)$$

where the individual reward components are defined as follows:

$$r_{\text{acc}}(o, y^*) = \mathbb{I}(\text{extract}(o) = y^*) \quad (6)$$

$$r_{\text{trace}}(o) = \alpha \cdot V_{\text{state}}(o) + (1 - \alpha) \cdot C_{\text{trans}}(o) \quad (7)$$

$$r_{\text{len}}(o) = \frac{2}{\pi} \arctan(k \cdot N_{\text{sent}}) \quad (8)$$

Here, $\mathbb{I}(\cdot)$ denotes the indicator function for factual correctness, and $r_{\text{trace}}(o)$ represents the structural reward derived from our metric (Section 3.2). We set $\lambda_{\text{acc}} = 2.0$ to prioritize accuracy. Crucially, for the auxiliary length reward $r_{\text{len}}(o)$, we utilize a scaling factor $k = 0.2$ to encourage a chain-of-thought (CoT) length exceeding 30 sentences. This ensures sufficient context for reliable TRACE extraction while enforcing comparable CoT lengths across settings

to serve as a control variable for verbosity. We also analyzed reward hacking behavior under different reward combinations, with details provided in Appendix D.3.

To rigorously evaluate TRACE, we compared the **Base model** (SFT) against two RL settings: a **Accuracy+Length** trained with only $r_{\text{acc}}$ and $r_{\text{len}}$, and **TRACE+Accuracy+Length** trained with the full reward including $r_{\text{trace}}$. By enforcing consistent CoT lengths across the latter two settings, we isolate the specific contribution of structural optimization from the effects of increased token generation. We utilized the GSM8K training set for RL fine-tuning, selected based on TRACE's relatively higher alignment with judge preferences in mathematical reasoning (Section 4.2). For evaluation, we reported performance on the GSM8K test set (in-distribution) and the ARC-Challenge test set (out-of-distribution); the latter was chosen because the domain analysis in Section 4.1 showed the largest performance gap between correct and incorrect answers. By observing performance changes when using TRACE as an RL reward signal, this setup allows us to verify consistency with the findings from Experiments 1 and 2. All RL experiments were implemented using the Transformer Reinforcement Learning (TRL) library (von Werra et al., 2020). Detailed training loss function and hyperparameters are provided in Appendix D.

*Table 6.* Performance comparison of DeepSeek-R1-Distill-Qwen-1.5B on GSM8K and ARC-Challenge using different rewards.

| Method | GSM8K | ARC-Challenge |
|---|---|---|
| Base Model (SFT) | 74.75% | 55.80% |
| Accuracy + Length | 77.86% | 57.93% |
| **TRACE + Accuracy + Length** | **84.69%** | **59.90%** |

**Results** As shown in Table 6, training with accuracy and length rewards alone (Accuracy + Length) yields measurable improvements over the Base model, confirming that encouraging longer reasoning provides some benefit. However, incorporating TRACE into the reward function produces substantially larger gains on both in-distribution (GSM8K) and out-of-distribution (ARC-Challenge) benchmarks.

These results suggest that incorporating structural guidance via TRACE provides complementary benefits beyond accuracy-based rewards. With the base model fixed and CoT length held constant across both RL settings, the performance gap indicates that TRACE can help guide models toward sounder reasoning processes rather than merely rewarding verbosity.

## 5. Limitations

TRACE evaluates argumentative structure and cognitive flow, not factual correctness. This design leads to inherent

failure modes. **False positives** occur when a model reasons fluently from an incorrect premise: the logical structure appears sound, but the conclusion is wrong due to factual errors, calculation mistakes, or misunderstanding of the question. **False negatives** arise when a model arrives at the correct answer through hesitant, poorly structured reasoning, such as lucky guesses, pattern matching, or memorization recall, which TRACE penalizes despite the correct outcome. We organize these failure modes into a four-quadrant taxonomy based on TRACE score (high/low) and answer correctness, with case studies provided in Appendix E.

The applicable scope of TRACE is shaped by these failure modes. The domain analysis in Section 4.1 shows that the gap between correct and incorrect TRACE values is larger in deductive domains and smaller in knowledge-retrieval domains, and the judge alignment in Section 4.2 shows higher prediction accuracy in math than in creative writing. Together, these observations suggest that TRACE is best used in reasoning-intensive settings where argumentative structure is informative, and applied with caution to tasks dominated by factual recall, code, or open-ended narrative.

Additionally, spaCy and TRACE-DeBERTa have limited robustness to mixed-format inputs such as code blocks, LaTeX equations, and narrative content, which can affect sentence segmentation and classification quality.

Finally, TRACE is ratio-based and does not account for reasoning length or intermediate step correctness. We plan to pursue improved methods that incorporate these factors in future work.

## 6. Conclusion

We introduced TRACE, a reference-free metric for evaluating LLM reasoning by analyzing Chain-of-Thought structure through Toulmin's argumentation framework. Our experiments across 7 models and 39 benchmarks show that TRACE correlates strongly with ground-truth accuracy ($r = 0.74$), outperforming surface-level metrics. TRACE also serves as an effective RL reward signal, providing complementary benefits beyond accuracy-only training.

We deliberately adopted rule-based scoring to prioritize interpretability, enabling direct inspection of penalized states and transitions. TRACE is not intended to replace existing evaluation methods. We expect that incorporating more sophisticated classifiers and additional humanities theories will further refine this framework. This line of research suggests that grounding LLM analysis in argumentation theory offers a promising direction toward explainable AI.

## Acknowledgments

This work has been supported by the Korea Institute of Science and Technology Information (grant K26L2M3C7).

## Impact Statement

This paper presents work whose goal is to advance the interpretability of LLM reasoning evaluation. We do not foresee direct negative societal consequences from this work.

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

# A. Experimental Details

## A.1. Model Snapshots

Table 7 summarizes the model versions used in our experiments. For the correlation analysis (Experiment 1), we used the latest available snapshots of each model. For Arena Hard v2.0 (Experiment 2), we used DeepSeek-R1 and QwQ-32B with GPT-4.1 as the judge. Finally, for the reinforcement learning experiments (Experiment 3), we utilized DeepSeek-R1-Distill-Qwen-1.5B.

*Table 7.* Model snapshots used in experiments.

| Model | Snapshot Date | Experiment |
|---|---|---|
| gpt-oss-120b | Aug 4, 2025 | Experiment 1 (Section 4.1) |
| gpt-oss-20b | Aug 4, 2025 | Experiment 1 (Section 4.1) |
| Claude-3.7-Sonnet | Feb 19, 2025 | Experiment 1 (Section 4.1) |
| Qwen-Turbo | Apr 28, 2025 | Experiment 1 (Section 4.1) |
| Qwen-Flash | Jul 28, 2025 | Experiment 1 (Section 4.1) |
| DeepSeek-R1 | May 28, 2025 | Experiment 1 (Section 4.1) |
| Kimi-K2-Thinking | Nov 4, 2025 | Experiment 1 (Section 4.1) |
| DeepSeek-R1 | Jan 20, 2025 | Experiment 2 (Section 4.2) |
| QwQ-32B | Mar 5, 2025 | Experiment 2 (Section 4.2) |
| GPT-4.1 (Judge) | Apr 14, 2025 | Experiment 2 (Section 4.2) |
| DeepSeek-R1-Distill-Qwen-1.5B | Jan 20, 2025 | Experiment 3 (Section 4.3) |

## A.2. Average Token Length per Model

Token counts were computed using the `tiktoken` library with the `cl100k_base` encoding. Table 8 reports the mean token length and mean sentences of reasoning blocks for each model.

*Table 8.* Mean token length and sentences of reasoning blocks per model.

| Model | Mean Tokens | Mean Sentences | Experiment |
|---|---|---|---|
| gpt-oss-120b | 503 | 30 | Experiment 1 (Section 4.1) |
| gpt-oss-20b | 846 | 55 | Experiment 1 (Section 4.1) |
| Claude-3.7-Sonnet | 1048 | 37 | Experiment 1 (Section 4.1) |
| Qwen-Turbo | 1616 | 94 | Experiment 1 (Section 4.1) |
| Qwen-Flash | 2198 | 111 | Experiment 1 (Section 4.1) |
| Kimi-K2-Thinking | 2257 | 117 | Experiment 1 (Section 4.1) |
| DeepSeek-R1 | 2773 | 124 | Experiment 1 (Section 4.1) |
| DeepSeek-R1 | 3903 | 197 | Experiment 2 (Section 4.2) |
| QwQ-32B | 5199 | 40 | Experiment 2 (Section 4.2) |

## A.3. Computational Cost Comparison

We compare the computational cost of TRACE against a representative 7B LLM judge in Table 9. The measurements use a single NVIDIA A100 GPU for GPU latency, and a standard CPU configuration for CPU-only latency.

*Table 9.* Computational cost comparison between TRACE and a 7B LLM judge per sample.

| | TRACE | LLM Judge (7B) |
|---|---|---|
| Model size | 184M | 7B |
| VRAM | $< 2$GB | $\sim$16GB |
| Latency (GPU) | $\sim$2s | $\sim$4s |
| Latency (CPU only) | $\sim$10s | Impractical |

## A.4. Domain Categorization

For the domain analysis in Section 4.1, we grouped the 39 benchmarks into 6 categories as shown in Table 10.

*Table 10.* Domain categorization of benchmarks.

| Domain | Benchmarks |
|---|---|
| Math & Logic | AIME24, AIME25
ARC Easy, Challenge
GSM8K
MMLU College Mathematics
MMLU-Pro Math |
| CS & Engineering | MMLU College Computer Science
MMLU-Pro Computer Science, Engineering
SuperGPQA Engineering |
| Natural Sciences | GPQA Main, Diamond
MMLU College Biology, College Chemistry, College Physics
MMLU-Pro Biology, Chemistry, Physics
SuperGPQA Science, Agronomy |
| Medicine & Health | MMLU College Medicine
MMLU-Pro Health, Other
SuperGPQA Medicine |
| Biz, Econ & Law | MMLU-Pro Business, Economics, Law
SuperGPQA Economics, Law, Management |
| Humanities & Social Sci. | MMLU-Pro History, Philosophy, Psychology
SuperGPQA Education, History, Literature & Arts, Philosophy |

## A.5. Baseline Implementation Details

We implemented three baseline metrics for comparison:

**Token Length.** We tokenized reasoning blocks using `tiktoken` with the `cl100k_base` encoding. In pairwise comparisons, the response with more tokens is predicted as the winner.

**Perplexity (PPL).** We computed perplexity using GPT-2 with a rolling window approach (context length = 1, max sequence length = 1024). In pairwise comparisons, lower perplexity is predicted as the winner.

**MTLD.** We calculated the Measure of Textual Lexical Diversity (MTLD) using the `lexicalrichness` library with default parameters. In pairwise comparisons, higher MTLD (more lexical diversity) is predicted as the winner.

## B. TRACE Framework Details

### B.1. TRACE-DeBERTa Training Data Details

TRACE-DeBERTa was fine-tuned from `microsoft/deberta-v3-base` on approximately 100K reasoning sentences. Labels were generated by prompting GPT-5.1 with detailed definitions and few-shot examples based on Toulmin's Argumentation Model and Flavell's Metacognition Theory.

---

**Label Definitions**

**Toulmin's Argumentation Model:**

- **Claim**: A conclusion, assertion, or answer being argued for.

- **Data/Evidence**: Concrete facts, observations, or given information.

- **Warrant**: Reasoning that connects evidence to claim.

- **Backing**: Support for the warrant (definitions, theorems, principles).

- **Qualifier**: Expressions of certainty or uncertainty.

- **Rebuttal**: Counterarguments, exceptions, or alternative considerations.

**Flavell's Metacognition Theory (extended):**

- **Monitoring**: Self-checking, tracking progress, noticing errors.

- **Evaluation**: Judging the quality or correctness of reasoning.

---

**Few-Shot Examples (Excerpt)**

**Mathematical Reasoning:**
*"By the tower law of field extensions, we have: $[K : F] = [K : E] \cdot [E : F]$."* → [Backing]
*"So the degree of the field extension is 4."* → [Claim]
*"Wait, I need to be more careful because $p = (1, 2, 5, 4)(2, 3)$."* → [Monitoring]

**Multi-Label Cases:**
*"$[\mathbb{Q}(\sqrt{2}) : \mathbb{Q}] = 2$ because $\sqrt{2}$ is irrational."* → [Evidence, Warrant]
*"The answer is (B), which is correct."* → [Claim, Evaluation]
*"I think most people would say that lying is wrong."* → [Claim, Qualifier, Backing]

**No Label Cases:**
*"Hmm."* → [ ]
*"Okay, let's tackle this question."* → [ ]
*"Thank you for listening."* → [ ]

---

### B.2. Allowed States

State Validity is computed based on the following set of allowed states, derived from valid combinations of Toulmin's components:

---

**Allowed States**

- Single labels: `Claim`, `Data/Evidence`, `Warrant`, `Backing`, `Evaluation`

- Composite labels: `Claim+Data/Evidence`, `Claim+Evaluation`, `Data/Evidence+Warrant`, `Warrant+Backing`, `Backing+Evaluation`

---

### B.2.1. LABEL DISTRIBUTION PER MODEL

Figure 6 illustrates the proportion of constructive elements, derived from 3.9K blocks per model in Section 4.1. We observe that the ratios of these elements within reasoning blocks vary significantly across models.

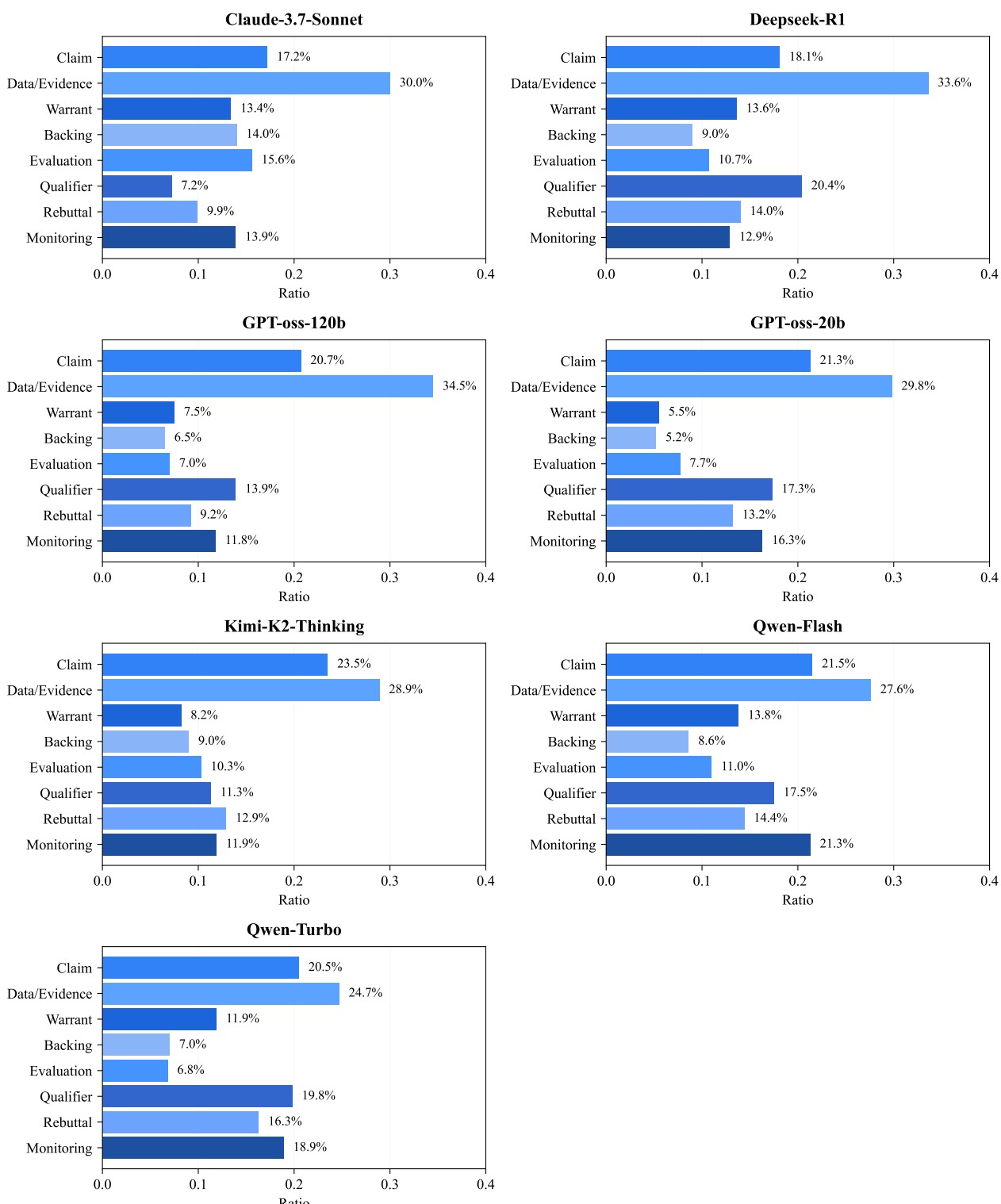

*Figure 6.* Distribution of constructive elements across models, derived from 26.3K reasoning blocks in Section 4.1.

## B.3. Good and Bad Transition Definitions

We systematically explored various transition set configurations. Initially, we tested three-way and four-way classifications (e.g., separating "moderately good" from "strongly good"), but these finer-grained distinctions introduced noise without improving correlation—the essential contrast between logical progression and cognitive stalling was already captured by the binary Good/Bad distinction.

To select the optimal transition sets, we enumerated all possible permutations of transition pairs and evaluated each configuration by its correlation with benchmark accuracy. The final sets reported here achieved the highest correlation among all tested configurations while remaining consistent with the theoretical principles of Toulmin's argumentation model and Flavell's metacognitive framework. Transitions not assigned to either set are treated as neutral. Transition Coherence is computed based on the following transition sets:

---

**Good Transitions**

Logical progressions that indicate sound argumentation flow:

| | |
|---|---|
| Data/Evidence $\rightarrow$ Claim | evidence-based conclusion |
| Data/Evidence $\rightarrow$ Warrant | explaining the evidence |
| Warrant $\rightarrow$ Claim | reasoning leads to conclusion |
| Claim $\rightarrow$ Backing | supporting the assertion |
| Claim $\rightarrow$ Evaluation | assessing the conclusion |
| Backing $\rightarrow$ Claim | grounded assertion |
| Backing $\rightarrow$ Evaluation | validating the support |
| Monitoring $\rightarrow$ {Claim, Data/Evidence, Backing, Evaluation} | productive self-correction |
| Rebuttal $\rightarrow$ {Claim, Backing} | resolving counterarguments |

---

**Bad Transitions**

Patterns indicating cognitive stalling or circular uncertainty:

| | |
|---|---|
| Monitoring $\leftrightarrow$ Monitoring | repetitive self-checking |
| Qualifier $\leftrightarrow$ Qualifier | compounding uncertainty |
| Monitoring $\leftrightarrow$ Qualifier | hesitation loops |
| Rebuttal $\leftrightarrow$ Rebuttal | unresolved contradictions |
| Rebuttal $\leftrightarrow$ Qualifier | uncertain counterarguments |

---

## C. Hyperparameter Selection

### C.1. Effect of $\alpha$ on Correlation and Accuracy

The weight $\alpha$ in the TRACE score formula balances State Validity and Transition Coherence. Figure 7 shows the effect of $\alpha$ on both Pearson correlation with benchmark accuracy and prediction accuracy on Arena Hard v2.0 (Math). We selected $\alpha = 0.7$ as it achieves near-optimal performance on both metrics.

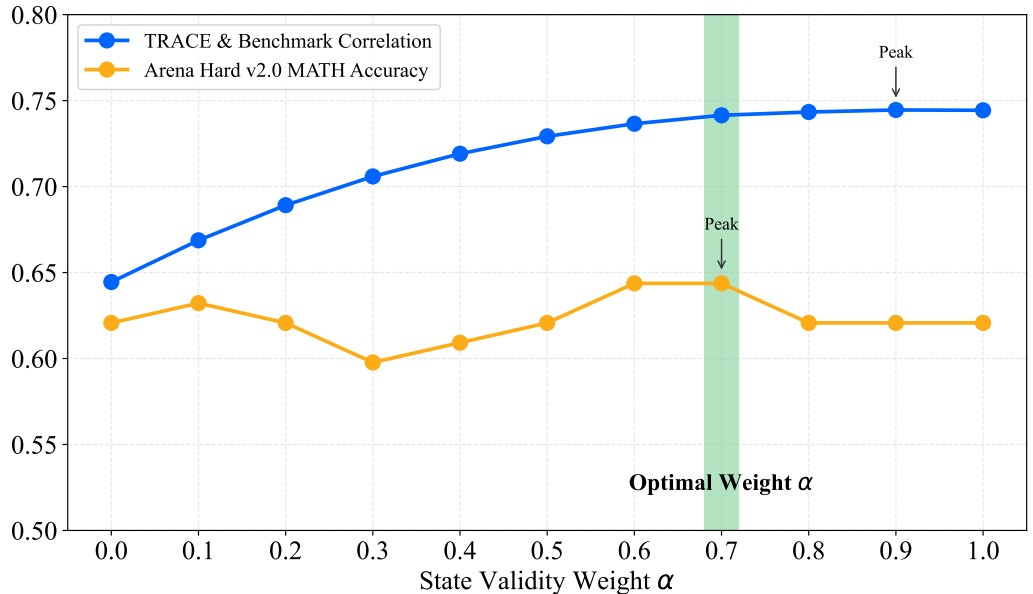

*Figure 7.* Effect of $\alpha$ on Pearson correlation with benchmark accuracy and prediction accuracy on Arena Hard v2.0 (Math). The shaded region indicates near-optimal performance.

### C.2. Statistical Significance

For the selected hyperparameter ($\alpha = 0.7$), we report the correlation coefficients with 95% confidence intervals computed via Fisher's z-transformation.

*Table 11.* Statistical significance of TRACE correlation at $\alpha = 0.7$.

| Metric | Coefficient | 95% CI | $p$-value |
|---|---|---|---|
| Pearson $r$ | 0.741 | [0.683, 0.791] | $< 0.001$ |
| Spearman $\rho$ | 0.755 | [0.699, 0.802] | $< 0.001$ |

All correlations are statistically significant at $p < 0.001$ across the full range of $\alpha$ values (see Appendix C.1).

## D. Reinforcement Learning Implementation Details

In Experiment 3, we utilized the **TRL (Transformer Reinforcement Learning)** framework to fine-tune the DeepSeek-R1-Distill-Qwen-1.5B model. This section details the reward formulation, the optimization objective, and the specific hyperparameters used in our experiments.

### D.1. Reward Formulation

As described in the main text, we employed a composite reward function $R(o, y^*)$ consisting of three components: accuracy, structural quality (TRACE), and length control. The precise mathematical formulation used in the training loop is as follows:

$$R(o, y^*) = \lambda_{\text{acc}} \cdot \mathbb{I}(\text{extract}(o) = y^*) + r_{\text{trace}}(o) + r_{\text{len}}(o) \tag{9}$$

where:

- **Accuracy Reward ($r_{\textbf{acc}}$):** A binary reward where $\mathbb{I}(\cdot)$ is 1 if the extracted answer matches the ground truth, and 0 otherwise. We applied a scaling weight of $\lambda_{\text{acc}} = 2.0$.

- **TRACE Reward ($r_{\textbf{trace}}$):** Computed as a weighted sum of State Validity ($V_{\text{state}}$) and Transition Coherence ($C_{\text{trans}}$) with $\alpha = 0.7$:
$$r_{\text{trace}}(o) = 0.7 \cdot V_{\text{state}}(o) + 0.3 \cdot C_{\text{trans}}(o)$$

- **Length Reward ($r_{\textbf{len}}$):** defined to prevent brevity bias while discouraging excessive verbosity:

$$r_{\text{len}}(o) = \frac{2}{\pi} \arctan(0.2 \cdot N_{\text{sent}})$$

  Here, $N_{\text{sent}}$ denotes the number of sentences in the reasoning chain, and the slope $k = 0.2$ was chosen to encourage chains of approximately 40 sentences.

### D.2. Optimization Objective (GRPO)

We optimized the policy $\pi_\theta$ using **Group Relative Policy Optimization (GRPO)**. For each prompt $q$, we sampled a group of $G = 4$ completions $\{o_1, \ldots, o_G\}$.

**Advantage Computation** To reduce variance and reward relative improvement, the advantage $\hat{A}_{i,t}$ for the $i$-th completion is computed by normalizing the total rewards within the group:

$$\hat{A}_i = \frac{R_i - \text{mean}(\{R_1, \ldots, R_G\})}{\text{std}(\{R_1, \ldots, R_G\}) + \epsilon} \tag{10}$$

where $R_i$ is the total reward for completion $o_i$.

**Loss Function** The model is trained using the clipped surrogate objective. Following recent best practices for reasoning models (e.g., DeepSeek-R1), we set the KL-divergence coefficient $\beta = 0$, relying on group-relative advantages for regularization. The loss function is defined as:

$$\mathcal{L}_{\text{GRPO}}(\theta) = -\frac{1}{G} \sum_{i=1}^{G} \frac{1}{|o_i|} \sum_{t=1}^{|o_i|} \min\left(\rho_{i,t} \hat{A}_i, \text{clip}(\rho_{i,t}, 1 - \epsilon, 1 + \epsilon)\hat{A}_i\right) \tag{11}$$

where $\rho_{i,t} = \frac{\pi_\theta(o_{i,t}|q, o_{i,<t})}{\pi_{\text{old}}(o_{i,t}|q, o_{i,<t})}$ is the probability ratio between the current and old policies, and $\epsilon = 0.2$ is the clipping range.

## D.3. Reward Hacking under Different Reward Combinations

During preliminary experiments, we observed reward hacking patterns under different reward combinations:

- **TRACE only:** The CoT length collapsed sharply, with outputs reduced to a single claim followed by minimal evidence and no hedging. A similar collapse has been reported in DAPO (Yu et al., 2026).

- **TRACE + Length:** Even with length controlled, the model produced well-structured but factually irrelevant reasoning, optimizing transition patterns without grounding.

- **TRACE + Accuracy + Length:** Adding the accuracy reward anchored factual grounding, while TRACE and length jointly shaped structural quality and verbosity. This combination mitigated the hacking patterns observed above.

## D.4. Hyperparameters

Table 12 lists the specific hyperparameters used for the GRPO training. All experiments were conducted on a single node with NVIDIA A100 GPUs using vLLM for accelerated generation.

*Table 12.* Hyperparameters for GRPO training on GSM8K.

| Hyperparameter | Value |
|---|---:|
| *Model & Data* | |
| Base Model | DeepSeek-R1-Distill-Qwen-1.5B |
| Dataset | GSM8K (Train Split) |
| *Training (GRPO)* | |
| Number of Generations ($G$) | 4 |
| Learning Rate | $1 \times 10^{-6}$ |
| Batch Size (per device) | 4 |
| Gradient Accumulation | 4 |
| Num Epochs | 1 |
| Optimizer | AdamW |
| Precision | bf16 |
| Clip Range ($\epsilon$) | 0.2 |
| KL Coefficient ($\beta$) | 0.0 |

# E. Qualitative Analysis

## E.1. Case Studies

To better understand TRACE's behavior, we categorize the possible outcomes into four quadrants based on TRACE score (High/Low) and answer correctness (Correct/Incorrect). Each quadrant contains distinct failure or success modes:

**High TRACE + Correct (Expected Positive)**

- *Normal case*: Sound argumentation structure with factually correct premises leading to correct conclusion.

- *Over-reasoning*: Excessive but well-structured reasoning that arrives at the correct answer.

**Low TRACE + Incorrect (Expected Negative)**

- *Normal case*: Poor structure combined with factual errors, leading to wrong conclusion.

- *Confused rambling*: Incoherent reasoning with no clear logical progression.

- *Circular reasoning*: Repetitive self-referential statements without advancing toward a conclusion.

**High TRACE + Incorrect (False Positive)**

- *Factual error with good structure*: Incorrect premise propagated through logically valid steps (see Appendix E.2).

- *Question misunderstanding*: Correctly structured reasoning applied to a misinterpreted problem.

- *Calculation error*: Sound reasoning with arithmetic mistakes in the final step.

- *Outdated knowledge*: Well-formed argument based on obsolete or incorrect information.

- *Wrong final selection*: Correct reasoning followed by selection of the wrong answer choice.

**Low TRACE + Correct (False Negative)**

- *Lucky guess with hesitation*: Uncertain, hesitant reasoning that coincidentally arrives at the correct answer (see Appendix E.3).

- *Pattern matching*: Direct keyword matching without explicit reasoning.

- *Memorization recall*: Retrieved answer from memory without constructing an argument.

- *Incomplete reasoning*: Abandoned logical reasoning in favor of intuition.

- *Self-doubt override*: Initially correct reasoning undermined by excessive self-questioning, yet returning to the original answer.

We present two representative case studies illustrating the most analytically interesting quadrants: **High TRACE + Incorrect** (Appendix E.2), which reveals the fundamental limitation that structural validity does not guarantee factual correctness, and **Low TRACE + Correct** (Appendix E.3), which demonstrates how TRACE can identify brittle reasoning even when outcomes happen to be correct.

## E.2. Case Study 1: False Positive

**High TRACE + Incorrect Answer**

**Model:** Claude-3.7-Sonnet
**Task:** Electrochemistry (GPQA)
**Question:** *"Thermodynamically, oxygen is a ...... oxidant in basic solutions. Kinetically, oxygen reacts ...... in acidic solutions. Which combination of weaker/stronger and faster/slower is correct?"*

**Ground Truth:** (C) weaker – slower     **Model Answer:** (D) weaker – faster
**TRACE Score:** 0.86     $(V_{state} = 0.90, \quad C_{trans} = 0.76)$

| Sentence | Label | $V_s$ | $\mathcal{T}_g$ | $\mathcal{T}_b$ | Valid |
|---|---|---|---|---|---|
| *Part 1: Thermodynamic Analysis (Correct)* | | | | | |
| This question asks about oxygen's behavior both thermodynamically and kinetically in different pH environments. | [Data] | 1.0 | – | – | – |
| In acidic solution: $O_2 + 4H^+ + 4e^- \rightarrow 2H_2O$, $E^\circ = +1.23$ V. In basic solution: $O_2 + 2H_2O + 4e^- \rightarrow 4OH^-$, $E^\circ = +0.40$ V. | [Data] | 1.0 | 0 | 0 | ✓ |
| Since the reduction potential is more positive in acidic solution (+1.23 V) than in basic solution (+0.40 V), oxygen is a stronger oxidant in acidic conditions. | [Data, Warrant] | 1.0 | 1 | 0 | ✓ |
| Therefore, in basic solutions, oxygen is a **weaker** oxidant. | [Claim] | 1.0 | 2 | 0 | ✓ |
| *Part 2: Kinetic Analysis (Incorrect Premise)* | | | | | |
| For the kinetics part, I need to think about the reaction rates of oxygen. | [Monitoring] | 0.0 | 0 | 0 | – |
| In general, protons ($H^+$) can catalyze many redox reactions involving oxygen. | [Backing] | 1.0 | 1 | 0 | ✗ |
| Acidic solutions have higher concentrations of $H^+$, which can facilitate the reduction of oxygen by providing protons needed in the reaction. | [Backing, Warrant] | 1.0 | 0 | 0 | ✗ |
| The reduction of oxygen in acidic solutions typically proceeds through multi-electron, multi-proton transfer steps, which can be accelerated by higher proton availability. | [Backing] | 1.0 | 0 | 0 | ✗ |
| Therefore, oxygen reactions are often **faster** in acidic solutions. | [Claim] | 1.0 | 1 | 0 | ✗ |
| *Part 3: Conclusion & Self-Verification* | | | | | |
| The answer that matches this combination is (D) weaker – faster. | [Claim, Evaluation] | 1.0 | 2 | 0 | ✗ |
| Actually, let me double-check my thermodynamic reasoning once more: The standard reduction potential for oxygen is indeed higher in acidic conditions than in basic conditions. | [Data, Monitoring] | 0.5 | 0 | 0 | – |
| This means oxygen has a greater tendency to gain electrons in acidic conditions, making it a stronger oxidant in acidic conditions. | [Backing, Warrant] | 1.0 | 2 | 0 | ✓ |
| Conversely, oxygen is a weaker oxidant in basic conditions. | [Claim] | 1.0 | 2 | 0 | ✓ |
| For kinetics, the presence of protons in acidic solutions generally accelerates oxygen reduction reactions. | [Backing] | 1.0 | 1 | 0 | ✗ |
| So my analysis confirms the answer: **(D) weaker – faster**. | [Claim, Evaluation] | 1.0 | 2 | 0 | ✗ |

**Analysis:** The model correctly applied thermodynamic reasoning using standard reduction potentials ($E^\circ$), arriving at the correct conclusion that oxygen is a *weaker* oxidant in basic solutions. However, the kinetic analysis contains a **factual error**: the assumption that "higher proton availability accelerates oxygen reduction" is an overgeneralization. In practice, oxygen reduction kinetics in acidic media are often *slower* due to higher overpotential barriers and different mechanistic pathways. Despite exhibiting well-formed argumentation structure with clear Evidence → Warrant → Claim progressions and productive self-verification (double-checking), the incorrect premise propagated through logically valid steps to a wrong conclusion. This case illustrates a fundamental limitation: **TRACE evaluates structural validity, not factual correctness**.

## E.3. Case Study 2: False Negative

---

**Low TRACE + Correct Answer**

**Model:** GPT-oss-20B
**Task:** Psychology (MMLU)
**Question:** *"A father brings his 22-year-old son for evaluation. Within the past 6 weeks, this formerly socially active young man has quit his job, has become withdrawn, and now spends most of his time in his room watching TV. He reports that he occasionally receives messages from aliens through the TV. What is the most appropriate DSM-IV diagnosis?"*

**Ground Truth:** (E) Schizophreniform Disorder  **Model Answer:** (E) Schizophreniform Disorder

**TRACE Score:** 0.42                        $(V_{state} = 0.37, \quad C_{trans} = 0.52)$

| Sentence | Label | $V_s$ | $\mathcal{T}_g$ | $\mathcal{T}_b$ | Valid |
|---|---|---|---|---|---|
| *Part 1: Initial Hesitation* | | | | | |
| We need to answer: DSM-IV diagnosis for 22-year-old with 6 weeks of symptoms... | [] | 0.0 | – | – | – |
| That suggests psychotic symptoms: auditory hallucinations? | [Monitoring, Rebuttal] | 0.0 | – | – | – |
| "Messages from aliens" could be auditory hallucination or delusion. | [Claim, Qualifier] | 0.5 | 2 | 2 | – |
| *Part 2: Reasoning with Uncertainty* | | | | | |
| DSM-IV: Schizophreniform disorder requires 1–6 months of schizophrenia-like symptoms. | [Backing] | 1.0 | 1 | 0 | ✓ |
| Brief reactive psychosis is for acute stressor, but no stressor mentioned. | [Rebuttal] | 0.0 | 0 | 0 | ✓ |
| So likely Schizophreniform Disorder. | [Claim, Qualifier] | 0.5 | 1 | 1 | – |
| Also could be Schizophrenia if >6 months, but only 6 weeks. | [Qualifier, Rebuttal] | 0.0 | 0 | 2 | ✓ |
| *Part 3: Uncertain Verification* | | | | | |
| So answer E. Let's double-check: Schizophreniform: 1–6 months of schizophrenia symptoms. | [Claim, Data, Monitoring] | 0.67 | 1 | 1 | ✓ |
| Yes. | [] | 0.0 | – | – | – |
| So answer is (E). | [Claim] | 1.0 | 2 | 0 | ✓ |

**Analysis:** Despite arriving at the correct answer, this reasoning trace exhibits significant structural weakness. The model shows persistent *cognitive stalling*, cycling between Qualifier and Rebuttal states (e.g., "could be..." → "Also could be..."). Although some good transitions occur ($\mathcal{T}_g = 7$), the $\mathcal{T}_{bad}$ count of 6 is notably high relative to the short reasoning chain, reflecting transitions such as [Claim, Qualifier] → [Qualifier, Rebuttal] that indicate uncertainty loops rather than logical progression. The low $V_{state}$ scores (averaging 0.37) further reveal that most sentences lack well-formed argumentative structure. The correct answer appears to result from domain knowledge recall rather than systematic deduction. This case demonstrates that **low TRACE scores can flag unreliable reasoning processes even when outcomes are correct**—useful for identifying brittle predictions that may fail under distribution shift.

