# OpenReview forum: "TRACE: Toulmin-based Reasoning Assessment through Constructive Elements for LLM CoT Evaluation"
_ICML.cc/2026/Conference — ICML 2026 regular_

### Official Review · Reviewer_e4tr · 2026-02-15

**Soundness:** 4
**Presentation:** 3
**Significance:** 3
**Originality:** 4
**Overall Recommendation:** 4
**Confidence:** 4

**Summary:**

This paper proposes TRACE, a reference-free metric for evaluating LLM chain-of-thought quality by labeling each sentence with Toulmin and metacognition elements, then scoring state validity and transition coherence via simple rules. The authors train a DeBERTa classifier for multi-label element tagging and evaluate TRACE on about 26K reasoning traces from 7 LLMs over 39 benchmarks, finding strong correlation with accuracy. They also test agreement with a GPT-4.1 judge on Arena Hard v2.0 and use TRACE as an RL reward on GSM8K.

**Compliance With Llm Reviewing Policy:**

Affirmed.

**Final Justification:**

The paper presents a clear and useful structure-aware metric for evaluating chain-of-thought quality, supported by broad experiments and promising RL results. My main concern was the reliability of the TRACE labels, and the rebuttal’s new human annotation study substantially addressed it. Some evaluation limitations remain (see my weaknesses above and my rebuttal acknowledgement below), but I maintain my weak accept recommendation.

**Key Questions For Authors:**

1. How reliable are the constructive element labels on real model outputs, as measured by human annotation on a small stratified sample across domains and formats?
2. Do you see evidence of reward hacking when optimizing rtrace, such as inflated “good” transitions without improved intermediate correctness. Is it possible to detect them automatically?
3. What is the per-example compute cost of TRACE scoring compared to an LLM judge?

**Limitations:**

Yes, the limitations are well discussed in Section 5 and Appendix E.

**Strengths And Weaknesses:**

The paper makes a decent case that structure-aware scoring can be useful beyond final-answer accuracy. It presents a clear and interpretable metric design that exposes which sentence types and element transitions drive the score.

Strengths:
- Solid empirical sweep across many benchmarks and multiple models, with sensible non-TRACE baselines for correlation.
- I like those practical RL experiment that controls for chain length and shows gains on GSM8K plus a small OOD transfer to ARC-Challenge.
- Good transparency on known failure modes, such as the concrete taxonomy and case studies that match the intended scope.
- Presentation is good with clear writing and many well-designed figures.

Weaknesses:
- TRACE-DeBERTa relies on LLM-generated labels (though this is type of approach is getting popular and recent works show that LLM-as-a-judge can achieve human-level capability), but there is no human-validated subset or agreement analysis to bound label noise.
- The allowed states and transition sets appear selected by searching permutations for best correlation, so a held-out selection protocol would reduce overfitting concerns.
- The judge-alignment study is narrow, using one judge model and one model pair, so it is hard to generalize beyond that setup. I guess a straightforward fix is to repeat the analysis with multiple independent judge LLMs and several diverse model pairs across different domains, and report agreement and ranking stability rather than a single judge-pair result.

---

> ### Author Rebuttal · Authors · 2026-03-30
>
> We sincerely thank the reviewer for the thorough and insightful review. The concern regarding label reliability is one we take seriously—we fully agree that the TRACE-DeBERTa classifier is the foundation of our entire pipeline, and that its validity directly shapes the credibility of all downstream conclusions, even if the downstream results appear encouraging. We have therefore conducted a human annotation study and provided details on our controlled data generation process to address this concern. We outline our responses below.
>
> &nbsp;
>
> > ### Weakness 1: Human validation of TRACE-DeBERTa labels
>
> Three senior NLP researchers independently annotated 400 sentences sampled from Experiment 1, balanced across models and label categories. Inter-annotator agreement reached a macro Cohen's κ of 0.672, reflecting the inherent difficulty of this fine-grained task. Using human labels as ground truth:
>
> | Label | Precision | Recall | F1-score |
> |---|:---:|:---:|:---:|
> | Claim | 0.696 | 0.634 | 0.662 |
> | Data/Evidence | 0.774 | 0.588 | 0.663 |
> | Warrant | 0.602 | 0.544 | 0.547 |
> | Backing | 0.780 | 0.612 | 0.685 |
> | Qualifier | 0.865 | 0.783 | 0.821 |
> | Rebuttal | 0.712 | 0.549 | 0.619 |
> | Monitoring | 0.803 | 0.585 | 0.675 |
> | Evaluation | 0.610 | 0.711 | 0.654 |
> | **Macro Avg** | **0.730** | **0.626** | **0.666** |
>
> These results will be included in the revised paper.
>
> &nbsp;
>
> > ### Weakness 2: Overfitting concerns in transition set selection
>
> We understand the concern. While the permutation search did converge on sets that align with Toulmin and Flavell's theoretical predictions, we acknowledge that a held-out validation protocol would have strengthened the claim. In future work, we plan to replace brute-force search with more principled methods (e.g., lightweight ML-based rule discovery) that are both effective and auditable.
>
> That said, per-model correlations remain consistent (r > 0.78) across all 7 models, indicating reasonable generalization:
>
> |LLM|Pearson r|n Benchmarks|
> |---|---|---|
> |gpt-oss-120b|0.8216|39|
> |gpt-oss-20b|0.8555|39|
> |claude-3.7-sonnet|0.8106|39|
> |qwen-turbo|0.8179|39|
> |qwen-flash|0.9144|39|
> |deepseek-r1|0.8340|39|
> |kimi-k2-thinking|0.7829|39|
>
>
> &nbsp;
>
> > ### Weakness 3: Narrow judge-alignment study
>
> We agree. The Arena Hard v2.0 experiment was limited to one judge and one model pair. The primary takeaway from this setup is not a general ranking claim, but rather identifying which domains TRACE is more informative in open-ended settings. However, as TRACE score distributions shift across models due to stylistic differences, a fair judge-alignment study would require careful control for model architecture, baseline accuracy, and score distribution—which involves a more extensive analysis than what could be addressed within the current scope. We plan to pursue this in future work.
>
>
> &nbsp;
>
> > ### Q1: Reliability of constructive element labels
>
> Please refer to our response to W1 above.
>
>
> &nbsp;
>
> > ### Q2: Reward hacking with r_trace
>
> Thank you for this question—we did in fact encounter this issue during experimentation.
>
> **(1) TRACE reward only (no length control):** CoT length collapsed dramatically (e.g., one claim + one evidence sentence with zero hedging). This is a known issue also reported in DAPO [1]. We addressed this by introducing a length-equalizing reward rather than distribution-matching.
>
> **(2) TRACE + length control (no accuracy):** Even with length controlled, reward hacking persisted exactly as the reviewer describes—the model produced well-structured but factually irrelevant transitions.
>
> **(3) TRACE + accuracy + length control:** Adding accuracy reward substantially mitigated this. For SFT-pretrained models, the RL signal targets alignment of state validity and transition coherence distributions while accuracy anchors factual grounding. In this setting, reward hacking becomes unlikely because structurally sound *but wrong* outputs are penalized.
>
> Regarding automatic detection: verifying the factual correctness of individual reasoning steps is beyond TRACE's scope (which prioritizes efficiency and simplicity). Approaches like PRM[2] that employ step-level LLM verification could complement TRACE for this purpose, but this remains future work. We will add discussion of these reward hacking dynamics to the revised manuscript.
>
> &nbsp;
>
> > ### Q3: Computational cost vs. LLM judge
>
> Thank you for raising this point. TRACE is complementary to LLM judges, not a replacement. Its key advantages are sentence-level diagnostics, zero configuration, and cost efficiency:
>
> | | TRACE | LLM Judge (7B) |
> |---|:---:|:---:|
> | **Model size** | 184M | 7B |
> | **VRAM** | < 2GB | ~16GB |
> | **with GPU per sample** | ~2 s | ~4 s |
> | **only CPU per sample** | ~10 s | Impractical |
>
> ---
> [1] Yu et al., DAPO: An Open-Source LLM Reinforcement Learning System at Scale, NeurIPS 2025.
>
> [2] Khalifa et al., Process Reward Models That Think, TMLR 2026.

---

> > ### Author Rebuttal · Reviewer_e4tr · 2026-04-02
> >
> > Thanks to the authors for the detailed rebuttal. My main concern was the reliability of the constructive-element labels underlying TRACE, and I think the new human annotation study substantially addresses that point. In particular, the added evaluation with three human annotators gives a much clearer sense of the classifier’s real-world reliability and strengthens confidence in the downstream analysis. The rebuttal also provides useful clarification on the RL setting: I appreciate the discussion of reward hacking, especially the fact that the authors explicitly observed failure modes when optimizing structural reward without sufficient grounding from accuracy. The response on efficiency is also helpful and makes the intended practical role of TRACE clearer.
> >
> > That said, I still view some of my earlier concerns as only partially resolved. In particular, the concern about selecting allowed states and transition sets based on correlation remains a methodological limitation, even if the per-model consistency is reassuring. Likewise, the judge-alignment experiment is still fairly narrow, and I do not think the rebuttal fully removes the concern about generality there.
> >
> > Overall, however, I believe the main questions I raised have been sufficiently addressed for my current score. The remaining issues now look more like scope and evaluation limitations than fatal flaws in the paper’s core contribution. I therefore keep my recommendation as weak accept and will be defending against rejection.

---

> > > ### Author Response · Authors · 2026-04-02
> > >
> > > Thank you for the thoughtful follow up and for confirming that the main concerns have been addressed. We greatly appreciate your engagement in the discussion.
> > >
> > > &ensp;Best regards,\
> > > &ensp;The Authors.

---

### Official Review · Reviewer_8otH · 2026-02-22

**Soundness:** 3
**Presentation:** 3
**Significance:** 3
**Originality:** 2
**Overall Recommendation:** 5
**Confidence:** 3

**Summary:**

This paper proposes TRACE, which scores and diagnoses models’ CoT reasoning by operationalizing Toulmin’s Argumentation Model and Flavell’s Metacognition Theory. Across multiple benchmarks, the authors show that TRACE correlates with model performance and judge preferences, and they further demonstrate its potential as a reward signal for improving reasoning during RL training.

**Compliance With Llm Reviewing Policy:**

Affirmed.

**Final Justification:**

The additional results are sufficient to support the main claims, so I am raising my score.

**Key Questions For Authors:**

Please refer to the weakness above.

**Limitations:**

yes

**Strengths And Weaknesses:**

Strengths:
1. Paper is well written and easy to follow.
2. TRACE provides an interpretable metric for CoT. To my understanding, once the sentence-level tagger is trained, TRACE scores can be computed offline, which can make it more scalable and potentially lower-cost.
3. The empirical evaluation is extensive, spanning many benchmarks and multiple LLMs.

Weakness:
My core concern is that the entire TRACE pipeline depends on the sentence-level labels produced by the tagger (TRACE-DeBERTa), which form the foundation of the method. Therefore, the reliability of these labels can substantially affect how solid the downstream experimental logic chain is. The training labels for the tagger are generated via GPT-5.1 annotation. While the classification metrics reported in Table 1 look reasonable, logically this mainly shows that the tagger can fit the distribution of GPT-generated labels in-distribution, and it does not directly guarantee that these labels remain semantically stable across different task domains or different CoT writing styles. To my reading, the authors’ additional justification of label reliability largely relies on a brief manual inspection (Lines 133–139), and I do not think this level of evidence is sufficient to directly support the robustness of downstream TRACE scores and subsequent conclusions. Similarly, Table 2 provides encouraging evidence that the overall TRACE score correlates with benchmark accuracy at an aggregate level. However, this correlation does not directly validate the semantic correctness or cross-domain stability of the underlying sentence-level labels, and the observed trend could still be driven by superficial style factors correlated with model strength. Targeted robustness checks on the tagger or sensitivity analyses would strengthen the causal link.

---

> ### Author Rebuttal · Authors · 2026-03-30
>
> We sincerely thank the reviewer for this critical observation. We fully agree that the classifier forms the foundation of the entire TRACE pipeline, and that its reliability directly determines the validity of all downstream conclusions—even if the downstream results appear encouraging. We believe that independent validation of the classifier is essential to the completeness of this work. Following your suggestion, we provide details on our controlled data generation process and report a human annotation alignment study below. These will be incorporated into the revised manuscript to strengthen the logical chain. We are grateful for the guidance that led to this important addition.
>
> &nbsp;
>
> > ### Q: Label reliability and downstream robustness
>
> To ensure semantic stability and stylistic diversity in training data, we took the following steps. Few-shot examples were drawn from publicly available governmental advisory committee transcripts and real LLM reasoning traces, then human-labeled. Domain-specific formats (e.g., LaTeX, chemical formulae) were manually crafted as additional shots. For the ~100K annotation generation, we alternated between GPT-5.1 and Claude 4.5 Sonnet to prevent single-model stylistic bias.
>
> We acknowledge, however, that in-distribution fit to LLM-generated labels does not constitute independent semantic validation. Ideally, the classifier should be verified with controlled variables such as domain, CoT style, and difficulty—but quantifying and grouping these factors is non-trivial. Our subsequent analyses (Domain Analysis, RL experiments) were designed to serve both as utility demonstrations and as indirect pipeline validation, but the rationale behind each was insufficiently articulated. We will make these motivations explicit in the revised manuscript.
>
> To directly address the need for independent validation, we conducted a human annotation study. Three senior NLP researchers independently labeled 400 sentences stratified across labels, domains, and models. Annotators were provided with guidelines grounded in Toulmin's and Flavell's definitions, and the use of LLMs was strictly prohibited during annotation. Inter-annotator agreement reached a macro **Cohen's κ of 0.672.** Using human labels as ground truth:
>
> | Label | Precision | Recall | F1-score |
> |---|:---:|:---:|:---:|
> | Claim | 0.696 | 0.634 | 0.662 |
> | Data/Evidence | 0.774 | 0.588 | 0.663 |
> | Warrant | 0.602 | 0.544 | 0.547 |
> | Backing | 0.780 | 0.612 | 0.685 |
> | Qualifier | 0.865 | 0.783 | 0.821 |
> | Rebuttal | 0.712 | 0.549 | 0.619 |
> | Monitoring | 0.803 | 0.585 | 0.675 |
> | Evaluation | 0.610 | 0.711 | 0.654 |
> | **Macro Avg** | **0.730** | **0.626** | **0.666** |
>
> Notably, the classifier's macro F1 of 0.666 approaches the inter-annotator agreement ceiling (κ = 0.672) on this 8-category multi-label task, suggesting that remaining errors largely reflect inherent task ambiguity rather than systematic classifier failure. The lowest-performing category, **Warrant (F1 = 0.547),** represents implicit inferential links between data and claims—a distinction that is inherently ambiguous even in human judgment, as such reasoning steps often overlap with adjacent categories. Conversely, categories with more explicit surface markers, such as **Qualifier (F1 = 0.821),** achieve substantially higher performance. We consider these per-category variations consistent with the expected difficulty gradient of the annotation taxonomy.
>
> These results, along with details on training data construction and inter-annotator agreement, will be included in the revised paper.

---

> > ### Author Rebuttal · Reviewer_8otH · 2026-04-03
> >
> > Thanks for the authors’ response. The additional results are sufficient to support the main claims, so I am raising my score.

---

> > > ### Author Response · Authors · 2026-04-03
> > >
> > > Thank you for raising your score. Your focus on classifier reliability pushed us to conduct the human annotation study, which we believe has meaningfully strengthened the paper. We are grateful for the guidance.
> > >
> > >  Best regards,\
> > >  The Authors.

---

### Official Review · Reviewer_284m · 2026-03-12

**Soundness:** 3
**Presentation:** 3
**Significance:** 3
**Originality:** 2
**Overall Recommendation:** 4
**Confidence:** 4

**Summary:**

This work presents TRACE, a novel reference-free metric for evaluating CoT in LLMs. By integrating Toulmin’s argumentation model and Flavell’s metacognitive framework, TRACE decomposes reasoning steps into constructive elements (Claim, Data, Warrant, ...) using a fine-tuned DeBERTa model. It then evaluates the quality of reasoning based on State Validity and Transition Coherence. The authors demonstrate that TRACE correlates strongly with benchmark accuracy across various models and tasks.

**Compliance With Llm Reviewing Policy:**

Affirmed.

**Key Questions For Authors:**

see weakness

**Limitations:**

yes

**Strengths And Weaknesses:**

Strengths:
The paper is well-structured and easy to follow. The narrative logic is clearly articulated.

Weaknesses:
Assessing the semantic validity of transitions between reasoning states is inherently difficult, arguably as challenging as Natural Language Inference (NLI) tasks. The paper validates the transition metric primarily through statistical correlation with the final answer correctness. This proxy validation raises concerns about the metric's discriminative power at the step level.

The proposed method models reasoning effectively as a linear chain of adjacent state transitions. Real-world argumentation often involves non-linear structures, such as multiple premises (A and B) jointly supporting a conclusion (C), or long-range dependencies where a later step resolves a contradiction introduced much earlier.

While the RL experiments show promise, the setting, fine-tuning a 1.5B model on GSM8K, is relatively limited. Demonstrating the effectiveness of TRACE in a "frontier" RL setting (e.g., larger models, more complex tasks) would make the claims significantly more solid.

The experimental comparison lacks strong modern baselines such as "LLM-as-a-Judge" (with or without rubrics). In Section 4.2, the paper uses an LLM judge as the ground truth, yet the agreement rate is moderate. If TRACE is less reliable than an LLM judge, its competitive advantage must lie elsewhere, presumably in cost or speed. However, the paper does not explicitly quantify this trade-off. Since TRACE requires sentence-level classification (using DeBERTa) for the entire chain, it is unclear how much more efficient it is compared to a single forward pass of a highly efficient LLM judge (e.g., GPT-4o-mini or a specialized 7B judge).

---

> ### Author Rebuttal · Authors · 2026-03-30
>
> We sincerely thank the reviewer for the detailed and constructive feedback. The suggestions on scaling RL experiments and clarifying efficiency trade-offs were particularly valuable in strengthening our work. We address each point below.
>
> &nbsp;
>
> > ### Q1: Linear chain assumption vs. non-linear argumentation structures
>
> We agree with this observation. Argumentation mining research on human texts does consider non-linear structures. However, we believe the linear chain modeling is reasonable for LLM reasoning blocks, for the following reasons.
>
> * **First,** LLM reasoning blocks average ~3,000 tokens—substantially shorter than completed human documents or scripts, limiting the scope for non-linear dependencies.
> * **Second,** LLM reasoning more closely resembles a think-aloud protocol[1] than a polished argumentative essay: thoughts unfold sequentially, making adjacent-state transitions a natural fit for this generation style.
> * **Third,** it is well-established that Transformer attention mechanisms are most heavily influenced by recently generated tokens[2], suggesting that adjacent transitions capture the most impactful dependencies in practice.
>
> That said, non-linear dependencies may still arise in LLM reasoning. We note that State Validity partially addresses this concern by evaluating each sentence's argumentative completeness independently of its position. Systematically analyzing whether LLM reasoning exhibits non-linear patterns is an interesting direction that we plan to explore in future work.
>
> &nbsp;
>
> > ### Q2: Scaling RL experiments to larger models and harder tasks
>
> We agree that the 1.5B experiment alone provides limited evidence. Following the reviewer's suggestion, we conducted an additional RL experiment using **Qwen-3-8B** with **LoRA-based** GRPO on LogiQA[3], a benchmark requiring multi-step deductive reasoning over complex logical passages—a substantially harder task than GSM8K's arithmetic chains.
>
> | Method | LogiQA |
> |---|:---:|
> | Base | 50.3% |
> | Accuracy + Length | 51.3% |
> | TRACE + Accuracy + Length | 53.2% |
>
> We note that LoRA updates only ~2% of total parameters, which inherently limits the magnitude of improvement. Nonetheless, the same directional pattern from the 1.5B experiment is reproduced: TRACE provides additive gains over accuracy+length rewards. We attribute this to TRACE encouraging coherent argumentative flow while suppressing repetitive overthinking loops (e.g., "But wait..." cycles) that are known to degrade reasoning performance[4].
>
> We will include these results and a clearer articulation of the RL experiments' purpose in the revised manuscript.
>
> &nbsp;
>
> > ### Q3: Comparison with LLM-as-a-Judge and efficiency trade-offs
>
> Thank you for this point. To clarify, TRACE is not intended to replace LLM-as-a-Judge but to serve as a **complementary** diagnostic tool. Its advantages lie in three areas:
>
> * **traceability**: LLM judges output a single score or preference, whereas TRACE diagnoses which reasoning has structural flaws—enabling targeted debugging.
> * **zero configuration**: LLM judges require per-task rubric design, output formatting, and prompt engineering. TRACE requires none—it can be applied to any CoT trace out of the box.
> * **cost efficiency**:
>
> || TRACE | LLM Judge (7B) |
> |---|:---:|:---:|
> | **Model size** | 184M | 7B |
> | **VRAM** | < 2GB | ~16GB |
> | **with GPU per sample** | ~2 s | ~4 s |
> | **only CPU per sample** | ~10 s | Impractical |
>
> TRACE does not capture factual correctness, but it provides a near-zero-cost structural health check. We plan to release TRACE as a pip-installable library with pretrained weights, scoring utilities, and transition heatmap visualizations (as in Figure 1).
>
> ---
> [1] Wei et al., Chain-of-Thought Prompting Elicits Reasoning in Large Language Models, NeurIPS 2022.
>
> [2] Liu et al., Lost in the Middle: How Language Models Use Long Contexts, TACL 2024.
>
> [3] Liu et al., LogiQA: A Challenge Dataset for Machine Reading Comprehension with Logical Reasoning, IJCAI 2020.
>
> [4] Chen et al., Stop Overthinking: A Survey on Efficient Reasoning for Large Language Models, TMLR 2025.

---

> > ### Author Rebuttal · Reviewer_284m · 2026-04-04
> >
> > Thanks for the response. My Q2 has been addressed. However, regarding Q1 and Q3, while I understand the authors' perspective and find their explanation reasonable, it may not be sufficient to completely resolve my concerns. This is not something that can be demonstrated through simple experiments. Overall, my attitude towards this work remains positive, and I will keep my score.

---

> > > ### Author Response · Authors · 2026-04-04
> > >
> > > We appreciate your continued engagement and are glad the additional RL results addressed your concern. We value your positive assessment.
> > >
> > >  Best regards,\
> > >  The Authors.

---

### Official Review · Reviewer_bM4Z · 2026-03-13

**Soundness:** 3
**Presentation:** 3
**Significance:** 3
**Originality:** 3
**Overall Recommendation:** 5
**Confidence:** 3

**Summary:**

The authors propose TRACE, a metric for evaluating reasoning quality beyond final answer accuracy. They argue that current evaluations focus only on correctness, ignoring whether the reasoning process itself is well-structured or coherent.
TRACE measures reasoning using two components: state validity and transition coherence. First, each sentence in a chain-of-thought trace is labeled using categories inspired by argumentation theory (claim, evidence, warrant, monitoring, etc.) via a trained classifier. Then, reasoning quality is scored based on whether individual steps form valid argumentative states and whether transitions between steps follow coherent patterns.
They evaluate TRACE across 7 models and 39 benchmarks, reporting a strong correlation (~0.74) with accuracy. They also show moderate alignment with LLM-as-a-judge preferences and demonstrate that adding TRACE as a reward signal in RL improves performance over accuracy+length rewards.

**Compliance With Llm Reviewing Policy:**

Affirmed.

**Final Justification:**

I think the rebuttal helped with the concerns a lot, those results and explanations need to be included in the main paper with the revision. I would also think about expanding the discussion section and talk about failed cases, etc there for a deeper insight. Overall I really enjoyed this paper and I think its a great contribution.

**Key Questions For Authors:**

I would like some explanation for groundedness and how Toulmin is relevant and generalizable, to look at some failure cases, and potentially learn about some controlled experiments. I am open to changing my mind positively on this paper.

**Limitations:**

Yes

**Strengths And Weaknesses:**

strengths:
- Clear framing of the problem which is timely and relevant
- Theory grounded design- connects COT reasoning (which is essentially arguments) to argument theory
- Process level evaluation of COT rather than just outcome level, the metric proposed is suggested to be used in conjunction with accuracy for better understanding reasoning, which is very sensible to me.
- 7 models, 39 benchmarks, 26K+ traces reported, very strong experimentation.
- Strong correlation with accuracy and interpretable results.

Weaknesses:
I was initially very impressed with the paper and results, but as I thought about it a bit more I found a few flaws that I couldn't get over.

- Even though it is grounded, there is no evidence provided for it. Math based reasoning can be different from code reasoning, which can follow execution not necessarily argumentation. Necessity, sufficiency or some evidence for why toulmin needs to be provided.
- Control for length, style is missing: as the authors also state TRACE rewards certainty and penalizes hedging, it rewards explicit claims, warrants, evaluations, which may just reward verbose reasoning which could be wrong.
- The classifier is trained on synthetic annotations produced by a strong LLM, yet no human-labeled validation set is provided. This creates a form of evaluation circularity: an LLM defines the notion of “good reasoning structure,” TRACE learns to replicate it, and the metric is then used to assess other LLMs. Without independent validation, it is unclear whether TRACE measures reasoning quality itself or stylistic similarity to the labeling model.
- To prove the high correlation number better, it needs to be controlled into difficulty, scale, style, etc. Stronger models are better at both writing and reasoning, easier problems produce cleaner traces, so these things need to be controlled for to understand that number further.
- Human eval would help to in this regard to show correlation between human preference.
- Failure modes need to be analyzed further: what are the cases where the trace score is high but answer is wrong? trace score is low but answer is correct? if its being introduced as a metric, we need to understand its limitations and use cases better

I would like some explanation for groundedness and how Toulmin is relevant and generalizable, to look at some failure cases, and potentially learn about some controlled experiments. I am open to changing my mind positively on this paper.

---

> ### Author Rebuttal · Authors · 2026-03-30
>
> We appreciate the reviewer for the insightful analysis; your perspectives have truly inspired us. While our experiments were conducted with a clear awareness of your concerns, we acknowledge the need to better explain their purpose and rationale. Below, we address your advice and outline our revisions.
>
> &nbsp;
>
> > ### Q1: Groundedness and relevance of Toulmin
>
> (1) First, TRACE analyzes the reasoning block, not the final output itself. Even in coding or math tasks, the model's internal deliberation ("Why this algorithm?", "Is this approach valid?") is fundamentally argumentative. Toulmin designed his model to capture exactly this: the universal process by which any claim becomes justified—from everyday reasoning ("Will it rain tomorrow?") to scientific and legal argumentation. The structure (Claim supported by Data through Warrant) is domain-agnostic.
>
> (2) Toulmin's model has traditionally been used to assess human argumentative writing, where structural adherence correlates with quality. **Does this extend to LLMs?** [1] showed that Toulmin-based SFT improved LLM argumentative writing aligned with human preference, and TRACE's transition analysis can naturally detect overthinking[2]. CoT is essentially "think-aloud" reasoning, where elements like Claim, Data, and Monitoring are clearly revealed—making argumentation theory a natural fit. We will strengthen the Introduction accordingly.
>
> &nbsp;
>
> > ### Q2: Control for length and style
>
> (1) This is why RL controls for both: CoT length is held constant via r_len, and the same base model controls for style. The performance gap between TRACE+Accuracy and Accuracy, with CoT length held constant, isolates TRACE's structural contribution. We will revise Sec 4.3 to frame this controlled aspect more explicitly.
>
> (2) We fully agree TRACE cannot detect factual errors—this is discussed in Limitations (Section 5) and Appendix E. However, TRACE is a lightweight structural health check (<2GB VRAM, ~2s per CoT, no task-specific setup), useful where ground-truth or LLM-judge setup is infeasible.
>
> &nbsp;
>
> > ### Q3: Evaluation circularity and lack of human validation
>
> (1) To mitigate stylistic bias in training data generation, we took several steps.
> **Few-Shot**: Few-shot examples were sampled from public transcripts and real LLM traces, then human-labeled; domain-specific cases were manually crafted.
> **Model**: For generation, we alternated between GPT-5.1 and Claude 4.5 Sonnet to ensure stylistic diversity across the ~100K annotations. We will make these details explicit in the revision.
>
> (2) We conducted a human evaluation study. Three senior NLP researchers independently annotated 400 sentences sampled from Exp 1, balanced across models and label categories. Inter-annotator agreement reached a macro **Cohen's κ of 0.672**, reflecting the inherent difficulty of this fine-grained annotation task. Using the human labels as ground truth, classifier achieved the following performance:
>
> |Label|Precision|Recall|F1-score|
> |---|:---:|:---:|:---:|
> |Claim|0.696|0.634|0.662|
> |Data/Evidence|0.774|0.588|0.663|
> |Warrant|0.602|0.544|0.547|
> |Backing|0.780|0.612|0.685|
> |Qualifier|0.865|0.783|0.821|
> |Rebuttal|0.712|0.549|0.619|
> |Monitoring|0.803|0.585|0.675|
> |Evaluation|0.610|0.711|0.654|
> |**Macro Avg**|**0.730**|**0.626**|**0.666**|
>
> These results will be included in the revised paper.
>
> &nbsp;
>
> > ### Q4: Understanding the correlation coefficient
>
> While defining quantifiable controls is challenging, we examined this through complementary analyses. **Domain:** The TRACE gap between correct and incorrect answers varies by field—being more pronounced in Natural Sciences than in Biz/Econ/Law—indicating sensitivity to reasoning type (Fig. 5). **Model:** Correlations remain strong (r > 0.78) across individual model styles (Tab. 4). **Length/style:** RL holds these constant, isolating TRACE's contribution (please see Q2). **Scale:** Per your suggestion, we compared TRACE scores for GPT-oss-120B vs. 20B:
>
> |Data|GPT-OSS-120B|GPT-OSS-20B|
> |---|:---:|:---:|
> |AIME|**.641**|.623|
> |GSM8K|**.751**|.686|
> |ARC|**.711**|.615|
> |MMLU|**.700**|.636|
> |MMLU-PRO|**.600**|.594|
> |GPQA|**.589**|.560|
> |SuperGPQA|**.568**|.537|
>
> The larger model consistently scores higher, as expected; its absence would have undermined TRACE's validity. We will articulate the rationale behind each analysis more clearly.
>
> &nbsp;
>
> > ### Q5: Human preference correlation
>
> As noted in Q1, [1] showed that Toulmin-structured outputs received higher human preference scores. While their setting differs from CoT, it supports the link between structured reasoning and preference. We plan to explore this in future work.
>
> &nbsp;
>
> > ### Q6: Failure mode analysis
>
> Sec 5 and App E already detail specific failure modes. However, to make the practical implications clearer, we will expand Section 5 with specific guidance on TRACE's applicable scope and caveats.
>
> ---
> [1] Xiao et al., Prove Your Point!, EMNLP 2024.
>
> [2] Chen et al., Overthinking, ICML 2025.

---

> > ### Author Rebuttal · Reviewer_bM4Z · 2026-04-03
> >
> > The authors addressed my concerns, and I am satisfied with the results/discussion. Appendix E is very convincing and needs to be referred to in the main paper more- additionally, Section 5 needs to be longer. Given those changes, I am happy to raise my score.

---

> > > ### Author Response · Authors · 2026-04-03
> > >
> > > Thank you for confirming that our responses have addressed your concerns. We will ensure that Appendix E is more prominently referenced in the main paper and expand Section 5 as suggested in the revised manuscript. We appreciate your constructive engagement throughout the review process.
> > >
> > >  Best regards,\
> > >  The Authors.

---

### Decision · Program_Chairs · 2026-04-30

**Decision:**

Accept (regular)

**Comment:**

This paper introduces TRACE, a reference-free metric for evaluating the quality of chain-of-thought (CoT) reasoning by operationalizing concepts from Toulmin’s argumentation theory and metacognitive frameworks. Across multiple reviewers, there is strong consensus that the paper addresses an important and timely problem—moving beyond outcome-based evaluation toward process-level assessment of reasoning in LLMs.

The problem formulation is clear and well-motivated: evaluating reasoning quality independently of final answer correctness is increasingly important for LLM research. TRACE offers an interpretable and structured framework (state validity and transition coherence), which reviewers found conceptually appealing and practically useful. The empirical evaluation is extensive (26K+ traces, multiple models and benchmarks), and consistently demonstrates strong correlation with accuracy. The method shows promise as a reward signal in RL, with consistent improvements over accuracy-only baselines. The paper is well-written and easy to follow.

The main concerns raised by reviewers focused on (1) the grounding and generality of the Toulmin-based formulation, (2) the reliability of the sentence-level classifier trained on LLM-generated labels, (3) potential confounding factors such as style/length and evaluation circularity, and (4) limitations in evaluation scope (e.g., judge comparisons, RL scale, transition modeling assumptions). The rebuttal effectively addressed a significant portion of these concerns.

Some concerns remain only partially addressed, though they are not deemed fatal: The theoretical justification for Toulmin-style structure as a universal proxy for reasoning quality, while plausible, is still not fully empirically validated across diverse reasoning paradigms. The reliance on linear transition modeling may miss non-local or compositional reasoning structures.  These are best viewed as scope limitations and future work directions, rather than flaws that undermine the core contribution.

Taking all reviews and the rebuttal into account, the paper makes a solid and timely contribution to LLM evaluation by proposing a interpretable and practical metric for reasoning quality. While some methodological and validation questions remain, the work is likely to stimulate further research and be useful to the community.